# Wildfire history of the boreal forest of southwestern Yakutia (Siberia) over the last two millennia documented by a lake-sedimentary charcoal record

Ramesh Glückler[1,2], Ulrike Herzschuh[1,2,3], Stefan Kruse[1], Andrei Andreev[1], Stuart Andrew Vyse[1], Bettina Winkler[1,4], Boris K. Biskaborn[1], Luidmila Pestrykova[5], and Elisabeth Dietze[1]

[1]Section of Polar Terrestrial Environmental Systems, Alfred Wegener Institute Helmholtz Centre for Polar and Marine Research, Potsdam, 14473, Germany
[2]Institute for Environmental Science and Geography, University of Potsdam, Potsdam, 14476, Germany
[3]Institute for Biochemistry and Biology, University of Potsdam, Potsdam, 14476, Germany
[4]Institute of Geosciences, University of Potsdam, Potsdam, 14476, Germany
[5]Institute of Natural Sciences, North-Eastern Federal University of Yakutsk, Yakutsk, 677007, Russia

*Correspondence to*: Ramesh Glückler (ramesh.glueckler@awi.de) and Elisabeth Dietze (elisabeth.dietze@awi.de)

**Abstract.** Wildfires, as a key disturbance in forest ecosystems, are shaping the world's boreal landscapes. Changes in fire regimes are closely linked to a wide array of environmental factors, such as vegetation composition, climate change, and human activity. Arctic and boreal regions and, in particular, Siberian boreal forests are experiencing rising air and ground temperatures with the subsequent degradation of permafrost soils, leading to shifts in tree cover and species composition. Compared to the boreal zones of North America or Europe, little is known about how such environmental changes might influence long-term fire regimes in Russia. The larch-dominated eastern Siberian deciduous boreal forests differ markedly from the composition of other boreal forests, yet data about past fire regimes remain sparse. Here, we present a high-resolution macroscopic charcoal record from lacustrine sediments of Lake Khamra (SW Yakutia, Siberia) spanning the last c. 2200 years, including information about charcoal particle sizes and morphotypes. Our results reveal a phase of increased charcoal accumulation between 600 and 900 CE, indicative of relatively high amounts of burnt biomass and high fire frequencies. This is followed by an almost 900-year-long period of low charcoal accumulation without significant peaks, likely corresponding to cooler climate conditions. After 1750 CE fire frequencies and the relative amount of biomass burnt start to increase again, coinciding with a warming climate and increased anthropogenic land development after Russian colonisation. In the 20th century, total charcoal accumulation decreases again to very low levels, despite higher fire frequency, potentially reflecting a change in fire management strategies and/or a shift of the fire regime towards more frequent, but smaller fires. A similar pattern for different charcoal morphotypes and comparison to a pollen and non-pollen palynomorph record from the same sediment core indicate that broad-scale changes in vegetation

composition were probably not a major driver of recorded fire regime changes. Instead, the fire regime of the last two millennia at Lake Khamra seems to be controlled mainly by a combination of short-term climate variability and anthropogenic fire ignition and suppression.

## 1 Introduction

Wildfires in Siberia have become larger and more frequent in recent years (Walker et al., 2019), drawing attention from both scientists and the wider public. The occurrence of wildfires in high latitudes is closely linked to a dry climate and to rising temperatures, which have been increasing at more than twice the rate of elsewhere (Jansen et al., 2020; Lenton, 2012). With ongoing warming, the established fire regimes will likely be subject to future change. Fire has long been the most important ecological disturbance in boreal forests, shaping the appearance and

composition of the world's largest terrestrial biome (Goldammer and Furyaev, 1996). It acts as a main driver behind the boreal forest's carbon pool, which comprises a third of all globally stored terrestrial carbon (Kuuluvainen and Gauthier, 2018) and strikes a fine balance between emitting carbon during combustion and sequestering it during following regrowth (Alexander et al., 2012; Ito, 2005; Köster et al., 2018). Wildfires are thought to turn this balance towards acting as a net source of emissions (Kelly et al., 2016; Walker et al., 2019),

while posing increasing risks to human livelihoods and infrastructure (Flannigan et al., 2009). Additionally, fire in boreal forests may trigger tipping points in tree mortality, tree density, and shifts in vegetation composition with continued global warming (Herzschuh, 2020; Lenton et al., 2008; Scheffer et al., 2012; Wang and Hausfather, 2020). For these reasons, a prediction of potential future changes in boreal fire regimes is imperative to inform and prepare adapted fire management strategies in a warming arctic and subarctic boreal. However, data about long-

term changes in fire regimes, a prerequisite for model validation, are still very sparse for large parts of the Russian boreal, despite it comprising more than half of the world's coniferous tree stocks (Nilsson and Shvidenko, 1998). Macroscopic charcoal particles in sediment archives, derived from biomass burning, are commonly used as a proxy for fire activity (e.g. Clark, 1988; Conedera et al., 2009; Remy et al., 2018; Whitlock and Larsen, 2001). Charcoal records are an important tool for tracking past changes in fire regimes and searching for underlying causes and

effects, from local to global scales (Marlon et al., 2008, 2013; Power et al., 2008). In recent years the evaluation of charcoal particle distributions was expanded to infer past fire frequencies (Higuera et al., 2007) and estimate source areas (Duffin et al., 2008; Leys et al., 2015), the type of vegetation burning with charcoal morphotypes (Enache and Cumming, 2007; Feurdean et al., 2020; Mustaphi and Pisaric, 2014), and fire intensity with charcoal reflectance (Hudspith et al., 2015). Yet, as the distribution of lake sediment studies in Russia is generally sparse

(Subetto et al. 2017), the Global Paleofire Database (www.paleofire.org; Power et al., 2010) lists only a few continuously sampled macroscopic charcoal records across the Siberian boreal forest. Only recently have charcoal records of sufficient temporal resolution allowed the assessment of fire return intervals in boreal European Russia and western Siberian evergreen forests, revealing close fire–vegetation relationships (Barhoumi et al., 2019; Feurdean et al., 2020). More studies have been conducted in North America (e.g. Frégeau et al., 2015; Hély et al.,

2010; Hoecker et al., 2020; Waito et al., 2018) and boreal Europe (e.g. Aakala et al., 2018; Feurdean et al., 2017; Molinari et al., 2020; Wallenius, 2011). However, comparisons across boreal study sites are complicated by the differing predominant fire regimes in North America (high-intensity crown fires) and Eurasia (lower-intensity surface fires) (de Groot et al., 2013; Rogers et al., 2015).

The main fire regimes in the European and western Siberian evergreen boreal forest also differ markedly from

those of its larch-dominated, deciduous counterpart in eastern Siberia. Many prevalent evergreen conifers (*Pinus sibirica*, *Picea obovata*, *Abies sibirica*) are commonly seen as fire avoiders and are more susceptible to crown fires (Dietze et al., 2020; Isaev et al., 2010; Rogers et al., 2015). The predominant eastern Siberian larches (*Larix gmelinii*, *L. cajanderi*, *L. sibirica*), on the other hand, can resist fires with an insulating bark protecting the cambium from heat, while their deciduous and self-pruning nature restricts fires from reaching the crown (Wirth,

2005). Moreover, larches are thought to benefit from occasional surface fires, leading to more saplings by clearing the lower vegetation layers of plant litter and mosses, although this might become a risk for young trees if fire frequencies increase above a certain threshold (Sofronov and Volokitina, 2010). Surface fires in larch forest might also play a role in the long-term preservation of permafrost (Dietze et al., 2020; Herzschuh, 2020; Herzschuh et al., 2016). These different fire strategies within the Siberian boreal forest reinforce the need for fire reconstructions

towards the eastern part to evaluate changes in fire regimes depending on the prevalent tree species and to obtain a biome-specific overview of fire regimes throughout time.

The main goal of this study is to start filling a pronounced gap in the global distribution of macroscopic charcoal records by providing the first continuously sampled, high-resolution macroscopic charcoal record from eastern Siberia, using charcoal size classes and morphotypes. We specifically aim to answer the following research

questions: (I) How did the fire regime in south-west Yakutia change throughout the last two millennia? (II) How might reconstructed fire history relate to common drivers behind changes in fire regimes (climate, vegetation, humans)?

## 2 Study site and methods

### 2.1 Location

Lake Khamra (59.97°N, 112.96°E) is located c. 30 km north-west of the Lena River in south-west Yakutia (Republic of Sakha, Lensky District), at an elevation of 340 m a.s.l. (Fig. 1a). Covering an area of 4.6 km² the lake has a maximum water depth of 22.3 m. The catchment area of 107.4 km² stretches around the gentle slopes surrounding the lake and extends in a south-western direction along the main inflow stream. It is embedded in a region of Cambrian bedrock made up of dolomite and limestone with transitions to silty Ordovician sandstone and locally restricted areas of clayey Silurian limestone (Chelnokova et al., 1988).

The surroundings of the lake were unglaciated during the Last Glacial Maximum (LGM; Ehlers and Gibbard, 2007) and discontinuous or sporadic permafrost can be present within the study area (Fedorov et al., 2018). Morphological features suggest that Lake Khamra is an intermontane basin lake that is not of thermokarst origin, because of a lack of steep slopes (Katamura et al., 2009a), a maximum lake depth of 22.3 m compared to thermokarst lakes reaching mostly <10 m (Bouchard et al., 2016; West and Plug, 2008), and the absence of cryogenic features such as ice wedges indicating thermokarst processes (Katamura et al., 2009b; Séjourné et al., 2015). Accordingly, the regional soil volumetric ice content as broadly defined by Fedorov et al. (2018) is probably below the 30% required for the development of thermokarst lakes (Grosse et al., 2013).

The region lies within the transition zone of evergreen to deciduous boreal forest. The lake catchment is covered by dense mixed-coniferous forest consisting predominantly of *Larix gmelinii*, together with *Pinus sylvestris, P. sibirica*, *Picea obovata*, *Abies sibirica*, *Betula pendula,* and *Salix* sp. (Kruse et al., 2019).

The continental climate of southern Yakutia is characterised by a short, mild growing season and extremely cold winters. The mean temperature of July at c. 18°C is contrasted by the low mean temperature in January of about -40°C (Fedorov et al., 2018). Mean annual precipitation from 2000–2016 at the closest weather station in Vitim (c. 60 km south-west of Lake Khamra) was c. 480 mm, with highest values in July to September (Russian Institute of Hydrometeorological Information, 2020). Wildfires most frequently occur in the central to southern regions of Yakutia, with varying stand-specific mean fire return intervals not exceeding 20–50 yrs at around 60°N, 120°E (Ivanova 1996) and 15 yrs near Yakutsk (Takahashi, 2006). Longer estimates, increasing with higher latitude, were found by Ponomarev et al. (2016) for central Siberia of 80 yrs at 62°N, 200 yrs at 66°N and 300 yrs at 71°N, with similar ranges reported by Kharuk et al. (2016).

The regional climate is mainly controlled by the large-scale Arctic Oscillation (AO) pattern emerging between cold high-latitude and mild mid-latitude air masses (Wu and Wang, 2002). It has been suggested that positive

phases of AO, with low atmospheric pressure over the Arctic directing warm air from the south, potentially increase Siberian fire activity (Balzter et al., 2005; Kim et al., 2020).

Overall population density of the Lensky District is rather low at 0.5 inhabitants per km² (Administrative Center Lensk, 2015). Lake Khamra is c. 40 km north of one of the larger district settlements, Peleduy, with a population of c. 5000 inhabitants. Traces of logging activity are visible on satellite imagery c. 10 km from the lake, while in its direct vicinity only winter forest tracks are kept open. We discovered traces of recent wildfire disturbance at vegetation plots around the lake, i.e. trees with fire scars on their trunk or burnt bark. These traces likely correspond

to recent fires within the catchment in 2006 and 2014 as captured by the remotely sensed forest loss data of Hansen et al. (2013) (Fig. 1b, c) and MODIS fire products (Giglio et al., 2018).

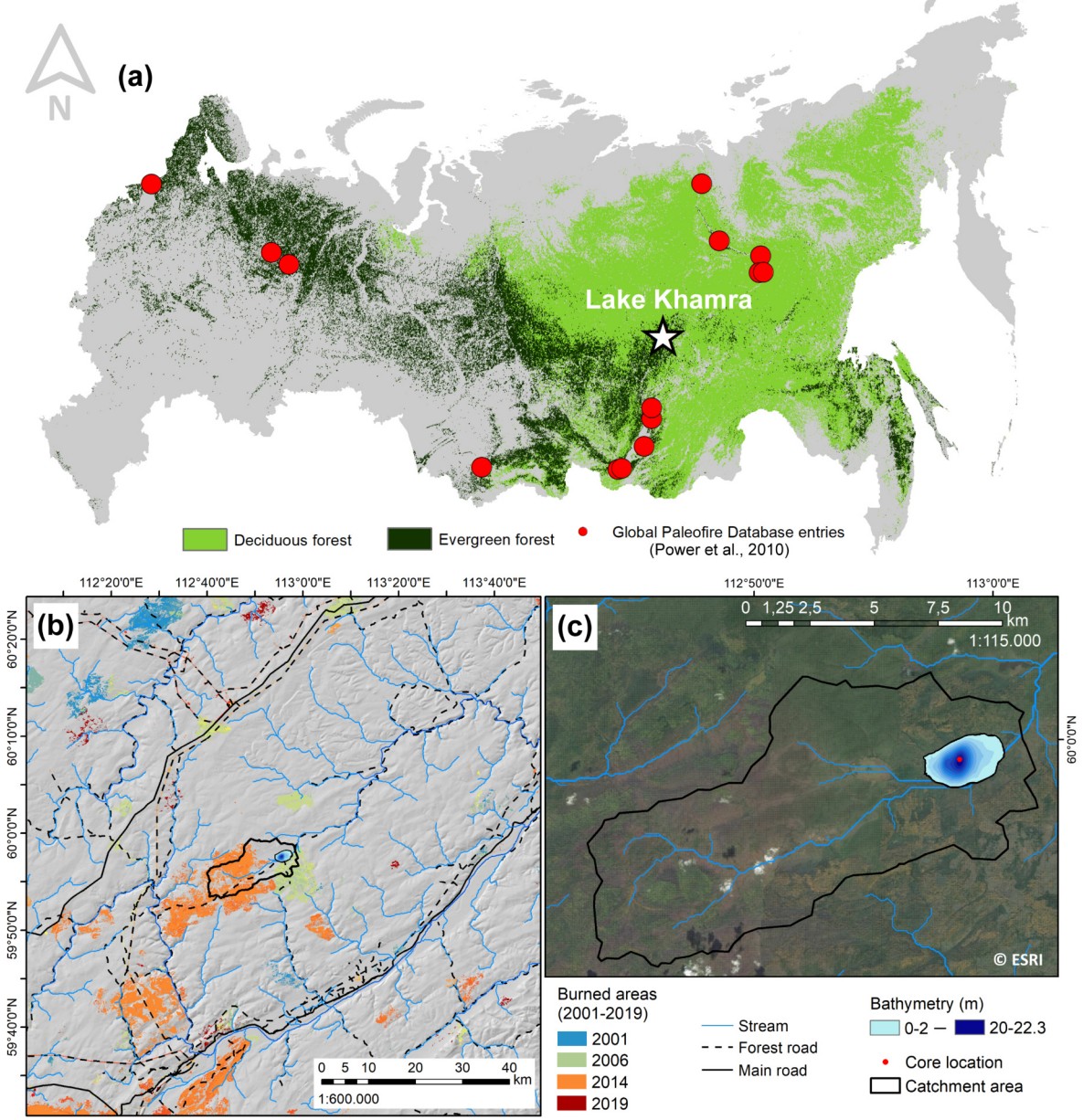

**Figure 1: (a) Position of Lake Khamra and existing Global Paleofire Database entries (accessed October 6, 2020) within the deciduous and evergreen boreal forests of Russia (land cover classification based on © ESA Climate Change Initiative Land cover project, provided via the Centre for Environmental Data Archival (CEDA). (b) Infrastructure and burned areas 2001–2019 CE (Giglio et al., 2018; Hansen et al., 2013) in the vicinity of the lake (centre). (c) Lake Khamra with bathymetry and catchment area/watershed (black line) (Service layer credits: Esri, DigitalGlobe, GeoEye, i-cubed, USDA, USGS, AEX, Getmapping, Aerogrid, IGN, IGP, swisstopo, and the GIS User Community).**


## 2.2 Fieldwork and subsampling

Fieldwork at Lake Khamra was conducted in August 2018. A 242-cm-long sediment core, EN18232-3, was retrieved using a hammer-modified UWITEC gravity corer at the deepest part of the lake (22.3 m), based on water depth measurements using a surveying rope and a hand-held HONDEX PS-7 LCD digital sounder. From the same location, a parallel short sediment core (EN18232-2, 39 cm length) was retrieved and subsampled in the field at increments of 0.5 cm (upper 20 cm) and 1 cm (lower 19 cm) for lead-210 and caesium-137 (Pb/Cs) age dating.

The sediment core, stored in a plastic tube, and all sediment samples, kept in Whirl-Pak bags, were shipped to the Alfred Wegener Institute, Helmholtz Centre for Polar and Marine Research (AWI) in Potsdam and placed in storage at 4°C. In October 2018, sediment core EN18232-3 was opened and cut in half in a cooled room under sterile conditions. While one half was archived, the other half was subsampled for proxy analysis and radiocarbon ([14]C) dating. The sampling scheme included one 2 mL sample from the midpoint of every 2 cm increment (n =

120), used for the combined palynological analysis and charcoal extraction, and two c. 1 mL samples between each pair of larger samples (n = 234), used specifically for charcoal extraction to ensure a continuous record of charcoal concentration. Bulk sediment samples for [14]C age dating were extracted every 20 cm (n = 12). At 85.5 cm core depth, a c. 2 cm long piece of wood was removed from the sediment core for [14]C dating. Due to a lack of any other larger organic structures, more macrofossil samples were picked while wet-sieving bulk sediment

samples (n = 15). Except for one case, a clear determination of their origin was not possible because of the small size of these samples.

## 2.3 Laboratory analyses

### 2.3.1 Lithology and age dating

Sediment core EN18232-3 was visually described before sampling. Water content and bulk density were

determined in subsamples from 120 sampling increments every 2 cm.

To establish a chronology, all bulk and macrofossil samples were sent to AWI Bremerhaven for accelerator mass spectrometry [14]C age dating at the MICADAS (Mini Carbon Dating System) laboratory. Subsamples of the parallel short core EN18232-2 (n = 19) were sent to the University of Liverpool Environmental Radioactivity Laboratory for Pb/Cs age dating, analysing [210]Pb, [226]Ra, [137]Cs, and [241]Am by direct gamma assay with Ortec HPGe GWL series

well-type coaxial low background intrinsic germanium detectors (Appleby et al., 1986). After careful evaluation of age dating results, an age-depth model was computed using Bacon v.2.4.3 (Blaauw and Christen, 2011; package

"rbacon"; Blaauw et al., 2020), combining Pb/Cs and adjusted $^{14}$C ages of all bulk samples calibrated with the IntCal20 $^{14}$C calibration curve (Reimer et al., 2020).

### 2.3.2 Macroscopic charcoal

We developed a sample preparation protocol that allows for the extraction of both macroscopic charcoal and the smaller pollen fraction including non-pollen palynomorphs (NPP) from the same sediment sample. *Lycopodium* tablets (Department of Geology, Lund University), used as marker grains in the palynological analysis, were dissolved in 10% HCl and added to the sediment samples. These were subsequently wet-sieved at 150 µm mesh width for separation of the macroscopic charcoal from smaller fractions (e.g. Conedera et al., 2009; Dietze et al.,

2019; Hawthorne et al., 2018). The suspension with the <150 µm-fraction was collected in a bowl below the sieve. This "pollen subsample" was then iteratively added to a falcon tube, centrifuged, and decanted prior to further preparation for palynological analysis. The >150 µm fraction in the sieve was rinsed together under a gentle stream of tap water before being transferred into another falcon tube. This macroscopic "charcoal subsample" was then left to soak overnight in c. 15 mL of bleach (<5% NaClO) to minimise the potential counting error from darker,

non-charcoal organic particles (Halsall et al., 2018; Hawthorne et al., 2018).

Counting of 304 charcoal samples was done under a reflected-light stereomicroscope at 10–40x magnification. All particles that appeared opaque and mostly jet-black with charred structures were counted in every given sample (see Brunelle and Anderson, 2003; Hawthorne et al., 2018). In addition, counted particles were grouped into three size classes (150–300, 300–500, and >500 µm measured along a particle's longest axis; Dietze et al., 2019) and

after similarities in shape (charcoal morphotypes; Enache and Cumming, 2007). For size reference, preparatory needles with known diameters of 300 and 500 µm were used that could be placed next to a charcoal particle. These needles also allowed the careful and non-destructive evaluation of the flexibility of particles of unknown origin, since charcoal fragments are described as fragile and non-bendable (Whitlock and Larsen, 2001).

Grouping of particles after their shape was based on the morphotype classification scheme by Enache and

Cumming (2007) and extended by three additional types to represent the variety of charcoal particles found at the study site. The original scheme differentiates between irregular (types M, P), angular (types S, B, C), and elongated (types D, F) shapes and further divides those depending on whether they show a visible structure or ramifications. The three additional types appear as highly irregular particles (type X), elongated, fibrous particles (type E), and slightly charred, partially transparent particles (type Z), the latter of which are not included in the total charcoal

sum. For correlations and visualisations, the morphotypes were grouped into their respective main categories (irregular, angular, or elongated). Within the topmost c. 50 cm of the sediment core, relative morphotype

distributions were retrospectively derived from counts of 67 subsamples. At that time, 11 samples distributed within the top 40 cm of the sediment core had already been used for other purposes and thus lack information on morphotype classification (total charcoal concentrations and size classes are available for all samples).

Eight randomly selected samples were counted a second time to obtain an estimate of counting uncertainty and ambiguity in charcoal identification.

### 2.3.3 Palynological samples

Established protocols following Andreev et al. (2012) were applied to the "pollen subsample" (n = 35). Of these samples, 11 were chosen specifically from intervals with high charcoal concentrations, whereas the others were

spread across the sediment core. Samples were treated with boiling potassium hydroxide for 10 min, sieved and left to soak in 18 mL hydrofluoric acid (40% HF) overnight. After two additional treatments with hot HF (1.5 h each), acetolysis was performed using acetic acid and in a second step a mixture of acetic anhydrite and sulfuric acid. After being fine-sieved in an ultrasonic bath, the samples were suspended in glycerol.

Pollen and NPPs were counted together with the added *Lycopodium* spores on pollen slides under a transmitting

light microscope, with a minimum count sum of 300 particles. For subsequent statistical analyses, relative frequencies of individual pollen taxa were calculated from the sum of terrestrial pollen. Spore, algal and non-pollen palynomorph percentages are based on the sum of pollen plus either spores, algae, or non-pollen palynomorphs, respectively (Andreev et al., 2020).

### 2.4 Statistical methods

Statistical analysis was carried out in R v.4.0.2 (R Core Team, 2020). To asses fire history, two different approaches were applied that decompose the charcoal record into a background component, representing long-term variations in charcoal accumulation and particle taphonomy, and a peak component, representing predominantly local charcoal accumulation during fire episodes (Higuera et al., 2007, 2009; Kelly et al., 2011; Whitlock and Anderson, 2003).

First, we ran a set of analyses referred to as "classic CHAR" (R script by Dietze et al., 2019), similar to the charcoal record decomposition in the well-established "CharAnalysis" (Higuera et al., 2009). We interpolated the charcoal record to equally spaced time intervals according to its median resolution and calculated the charcoal accumulation rate (CHAR, particles $cm^{-2}$ $yr^{-1}$; package "paleofire" v.1.2.4, function "pretreatment"; Blarquez et al., 2014). A background component was determined by computing a locally estimated scatterplot smoothing (LOESS) at a

window width of 25% of the total record length (package "locfit" v.1.5-9.4, function "locfit"; Loader et al., 2020).

This window width was found to result in an efficient distribution of a signal-to-noise index (SNI) >3 after Kelly et al. (2011), which indicates a high degree of separation between signal and noise (Barhoumi et al., 2019; Kelly et al., 2011). A peak component was created by subtracting the background component from the timeseries. By fitting two Gaussian distributions into the histogram of the peak component (package "mixtools" v.1.2.0, function "normalmixEM"; Benaglia et al., 2009), a global threshold was defined at the 99th percentile of the noise distribution (Gavin et al., 2006; Higuera et al., 2011). All peak component values exceeding the threshold were subsequently identified as signal (representing fire episodes) and marked when they overlapped with periods of SNI >3, indicating a clear distinction from surrounding noise. Usually in instances of multiple consecutive samples, only the highest peak above threshold is recognised, but we included all of them as fire episodes in this case. Recent fires within the lake's catchment that were just 8 yrs apart (Fig. 1b), as well as a predominantly atmospheric input of charcoal and a quick recovery of the filtering wetland vegetation around the large lake, led to the suspicion that the ability of a single fire to create high charcoal counts in multiple samples is limited at this site. However, an absolute minimum estimate of the number of fire episodes was obtained by considering only one fire episode from a given peak in CHAR, and only those that are clearly separated from noise (i.e. overlapping with phases of SNI >3). A global threshold was chosen to fit the steady sedimentation rate and vegetation composition around the lake. We also tested a local threshold to see how robust our approach is when compared to alternatives. For this, Gaussian distributions were fitted to samples of the peak component within a moving window of the same width as used for background component determination (package "zoo"; function "rollapply"; Zeileis and Grothendieck, 2005). Subsequent threshold values were obtained the same way as before, but additionally smoothed with a default LOESS to minimise the impact of outliers.

Fire return intervals (FRIs) were determined as the temporal difference between subsequent fire episodes. An illustration of fire frequency, as the rate of identified fire episodes, was derived by counting all fire episodes within a moving window spanning 200 yrs before applying a LOESS of the same window width to provide a clearer visualisation.

To account for accumulated uncertainties from both the chronology and the counting procedure, we used a Monte Carlo (MC) based approach (for detailed description and R script see Dietze et al., 2019), referred to as "robust CHAR". In short, it describes both the age and proxy values of each sample as Gaussian distributions, creating a pool from which random values are sampled. As inputs we used the 2σ range of Pb/Cs ages and 1σ range of calibrated and adjusted $^{14}$C ages to scale the general magnitudes of uncertainty between the two dating methods to comparable dimensions. For proxy uncertainty, the average deviation between the repeatedly counted samples was used (c. 20%). Following Dietze et al. (2019), we aggregated three consecutive samples to scale the high resolution

of the charcoal record to the relatively high age and counting uncertainties. Five thousand MC runs were performed and output data resolution set at three times the record's median temporal resolution (18 yrs). Robust CHAR was then divided into background and peak components, similarly to classic CHAR, but by computing 1000 randomly sampled LOESS fits at varying window widths ranging from 5–25% of the record length, thereby not relying on an individual user-input value (Blarquez et al., 2013).

The statistical approach outlined above (classic and robust CHAR including the determination of SNI, FRIs, and fire frequency) was also applied to the charcoal size classes and morphotype categories to assess their individual contribution to the sum of all particles, whether different charcoal groups represent different types of fires, the potential relationships between charcoal particle size and source area, and/or varying fuel types over time (see Supplement).

A principal component analysis (PCA; package "stats"; R Core Team, 2020) was used to assess relationships among the various centred log-ratio (clr) transformed (package "compositions"; van den Boogaart et al., 2020) relative distributions of charcoal particle classes. Assuming that the various charcoal morphotypes were formed by different types of vegetation burning, we would expect an increase of a specific morphotype to coincide with changes in the distribution of some plant types in the vegetation composition, also represented by the pollen spectra. To explore this hypothesis, we applied a correlation test using Kendall's $\tau$ (package "psych"; Revelle, 2020) to clr-transformed relative distributions of pollen groups that were expected to be impacted by wildfires (pollen sums of arboreal, non-arboreal, deciduous and evergreen taxa) and charcoal classes following Dietze et al. (2020).

## 3. Results

### 3.1 Lithological sediment properties and chronology

Sediment core EN18232-3 shows no lamination or visible changes in its brown colour or the texture of the sediment matrix. Its homogenous appearance is underlined by both uniform mean dry bulk density (184 ± 4 mg cm$^{-3}$, mean ± 1$\sigma$) and water content (83 ± 4%).

Bulk sediment $^{14}$C ages indicate a rather linear chronology, with only the deepest two samples returning similar ages. In contrast to this, the $^{14}$C ages of the macrofossils do not form a clear chronological pattern and are not in good agreement with the bulk sediment samples. Only 9 of the 15 macrofossil samples were large enough to be used for $^{14}$C dating, and mostly only just above the minimum amount of carbon (Table 1a). As Pb/Cs dating from the parallel short core EN18232-2 reveals no disturbances in its uniform sedimentation rate, the ages are expected

to be applicable to the main core EN18232-3 from the same location within the lake and thus can be compared to the respective [14]C ages. Noticeably, the topmost [14]C bulk sample dates to 1415 ± 27 [14]C yrs BP (before present, i.e. before 1950 CE), whereas the Pb/Cs method from the parallel core confirms the expected recent surface age (Table 1b). This [14]C age offset can have a variety of causes, such as sediment mixing processes (Biskaborn et al.,

2012), the presence of old organic carbon (Colman et al., 1996; Vyse et al., 2020), or dissolved carbonate rock (hard-water effect; Keaveney and Reimer, 2012; Philippsen, 2013).

In the case of Lake Khamra, located in a zone of discontinuous permafrost and bedrock containing early Palaeozoic carbonates, the observed [14]C age offset is a likely consequence of input of both old organic and inorganic carbon through the south-western inflow stream. The magnitude of this offset is documented by the difference between

the [14]C bulk surface sample and the corresponding Pb/Cs age. There are indicators to support the assumption that accumulation of old carbon not only happened recently, but rather constitutes an ongoing process in this lake system. First, treating the surface [14]C age as an outlier without also adjusting the other [14]C dates would necessarily lead to a strong shift in sedimentation rates, which is neither reflected by the homogenous appearance and density, nor the uniform sedimentation rate as implied by the Pb/Cs method. This is underlined by reports of stable lake

conditions and thermokarst processes in central Yakutia during the late Holocene (e.g. Pestryakova et al., 2012; Ulrich et al., 2019). Second, macrofossil [14]C ages older than the surrounding sediment matrix provide direct evidence for the potential influence of old carbon on bulk sediment samples at various depths (e.g. macrofossil age of 9902 ± 97 [14]C yrs BP in sediment that dates back to only c. 100 yrs BP according to the parallel core's Pb/Cs age). For these reasons, the documented age offset (mean ± 1σ) in the topmost [14]C sample was used to adjust

all other [14]C bulk samples (Colman et al., 1996; Vyse et al., 2020). Although macrofossil samples are usually thought to be superior in precision to bulk sediment for age-dating purposes (Hajdas et al., 1995; Wohlfarth et al., 1998) and have also been used to quantify a [14]C age offset throughout a sediment core (Gaglioti et al., 2014), within the present environment we cannot exclude a potential permafrost origin as well as measurement uncertainties due to their small size. For this reason, they were not used for constructing the chronology.

The age-depth model created according to these observations (Fig. 2) shows a smooth transition from Pb/Cs dates towards adjusted bulk [14]C ages. Its uniform sedimentation rate mirrors the sediment core's homogenous composition and supports the underlying assumption of a rather constant magnitude of old carbon influence on bulk [14]C ages. Based on this chronology, the sediment core spans c. 2350 years across its 242 cm length. The continuously sampled charcoal record spans c. 2160 years and thus reaches back from 2018 CE until c. 140 BCE.

Mean (± 1σ) sampling resolution of the whole record is 7.1 ± 4.1 yrs (max: 27, min: 3), including the pollen samples, which cover on average 8.9 ± 3.8 yrs (max: 25, min: 4).

| Lab-ID | Depth (cm) | Type | $F^{14}C \pm 1\sigma$ | $^{14}C$ BP $\pm$ $1\sigma$ | cal BP $\pm 2\sigma$ | cal BP $\pm 2\sigma$ (adjusted) | Comment |
|---|---|---|---|---|---|---|---|
| **3746.1.1*** | **0.5** | **Bulk** | **0.8385 ± 0.0029** | **1415 ± 27** | **1321.5 ± 32.5** | **-** | **$^{14}C$ age-offset** |
| 3733.1.1 | 20–21 | Macrofossil | 0.8876 ± 0.0119 | 958 ± 108 | 924.5 ± 247.5 | - | Unknown source, 18 µg C |
| 3734.1.1 | 21–22 | Macrofossil | 0.2915 ± 0.0035 | 9902 ± 97 | 11459 ± 277 | - | Unknown source, 127 µg C |
| 3735.1.1 | 25–26 | Macrofossil | 0.7669 ± 0.0072 | 2132 ± 75 | 2132.5 ± 191.5 | - | Unknown source, 144 µg C |
| 3736.1.1 | 29–31 | Macrofossil | 0.9069 ± 0.0081 | 785 ± 72 | 733 ± 170 | - | Birch seed, 32 µg C |
| 3737.1.1 | 30–31 | Macrofossil | 0.9473 ± 0.0090 | 435 ± 76 | 466 ± 156 | - | Unknown source, 36 µg C |
| **3747.1.1*** | **35–37** | **Bulk** | **0.8054 ± 0.0028** | **1738 ± 28** | **1639 ± 70** | **385 ± 79** | |
| 3738.1.1 | 49–50 | Macrofossil | 0.7719 ± 0.0077 | 2080 ± 80 | 2069 ± 237 | - | Unknown source, 56 µg C |
| 3739.1.1 | 49–51 | Macrofossil | 0.7720 ± 0.0098 | 2079 ± 102 | 2077 ± 254 | - | Unknown source, 22 µg C |
| **3748.1.1*** | **77–79** | **Bulk** | **0.7287 ± 0.0030** | **2542 ± 33** | **2622 ± 127** | **1067 ± 106** | |
| 3741.1.1 | 85.5 | Macrofossil | 0.8566 ± 0.0031 | 1244 ± 29 | 1172 ± 100 | - | Piece of wood, 986 µg C |
| 3742.1.1 | 97–99 | Macrofossil | 0.7799 ± 0.0074 | 1997 ± 76 | 1930 ± 191 | - | Unknown source, 35 µg C |
| **3749.1.1*** | **117–119** | **Bulk** | **0.7164 ± 0.0025** | **2679 ± 28** | **2797.5 ± 46.5** | **1187 ± 96** | |
| **3750.1.1*** | **157–159** | **Bulk** | **0.6773 ± 0.0027** | **3130 ± 32** | **3345.5 ± 97.5** | **1628.5 ± 72.5** | |
| **3751.1.1*** | **197–199** | **Bulk** | **0.6417 ± 0.0026** | **3564 ± 32** | **3847.5 ± 121.5** | **2156 ± 148** | |
| **3752.1.1*** | **237–239** | **Bulk** | **0.6427 ± 0.0023** | **3551 ± 28** | **3822 ± 99** | **2152.5 ± 146.5** | |

**Table 1a: $^{14}C$ age results of bulk and macrofossil samples from core EN18232-3. All samples used for the chronology are marked with a * in the first column. Lab-IDs 3732.1.1, 3740.1.1, 3743.1.1, 3743.2.1, 3745.1.1 and 3753.1.1 (not shown) belong to the 6 macrofossil samples too small to be successfully dated.**


| Depth (cm) | $^{210}$Pb ± 1σ (total, Bq kg$^{-1}$) | $^{210}$Pb ± 1σ (unsupported, Bq kg$^{-1}$) | $^{137}$Cs ± 1σ (Bq kg$^{-1}$) | Age BP ± 1σ | Year CE ± 1σ |
|---|---|---|---|---|---|
| 0 | - | - | - | -68 ± 0 | 2018 ± 0 |
| 0.5–1 | 1707.8 ± 38.5 | 1664.7 ± 38.9 | 38.7 ± 4.1 | -64 ± 0 | 2014 ± 0 |
| 1.5–2 | 1382.2 ± 33.3 | 1340.7 ± 33.8 | 43.4 ± 4.3 | -58 ± 1 | 2008 ± 1 |
| 2.5–3 | 1096.4 ± 32.4 | 1046.9 ± 32.8 | 54.0 ± 3.7 | -52 ± 1 | 2002 ± 1 |
| 3.5–4 | 817.8 ± 24.7 | 780.6 ± 25.0 | 61.9 ± 3.8 | -46 ± 2 | 1996 ± 2 |
| 4.5–5 | 507.4 ± 18.6 | 476.8 ± 18.9 | 106.7 ± 3.5 | -39 ± 2 | 1989 ± 2 |
| 5.5–6 | 505.2 ± 22.5 | 467.4 ± 22.9 | 118.7 ± 4.9 | -33 ± 3 | 1983 ± 3 |
| 6.5–7 | 464.8 ± 14.6 | 427.2 ± 14.9 | 144.7 ± 3.3 | -26 ± 4 | 1976 ± 4 |
| 7.5–8 | 316.2 ± 19.3 | 284.6 ± 19.6 | 216.8 ± 5.6 | -20 ± 4 | 1970 ± 4 |
| 8.5–9 | 262.8 ± 14.6 | 222.6 ± 15.0 | 286.1 ± 4.6 | -13 ± 5 | 1963 ± 5 |
| 9.5–10 | 254.6 ± 11.5 | 216.1 ± 11.8 | 136.2 ± 3.2 | -6 ± 5 | 1956 ± 5 |
| 10.5–11 | 254.8 ± 14.7 | 220.6 ± 14.9 | 47.8 ± 2.6 | 1 ± 5 | 1949 ± 5 |
| 11.5–12 | 152.7 ± 13.8 | 114.8 ± 14.2 | 19.4 ± 2.6 | 8 ± 5 | 1942 ± 5 |
| 12.5–13 | 148.0 ± 9.8 | 106.0 ± 10.0 | 12.7 ± 1.3 | 16 ± 6 | 1934 ± 6 |
| 13.5–14 | 110.5 ± 7.0 | 73.7 ± 7.2 | 9.6 ± 1.3 | 23 ± 7 | 1927 ± 7 |
| 14.5–15 | 89.1 ± 7.7 | 57.8 ± 7.9 | 5.2 ± 0.9 | 31 ± 8 | 1919 ± 8 |
| 16–17 | 81.7 ± 8.8 | 45.7 ± 9.1 | 2.2 ± 1.4 | 45 ± 10 | 1905 ± 10 |
| 18–19 | 68.6 ± 9.7 | 26.0 ± 10.0 | 2.0 ± 1.4 | 61 ± 13 | 1889 ± 13 |
| 20–21 | 47.8 ± 6.4 | 11.9 ± 6.6 | 1.0 ± 1.0 | 78 ± 16 | 1872 ± 16 |
| 22–23 | 47.3 ± 7.6 | 8.2 ± 7.9 | 1.8 ± 1.1 | 95 ± 19 | 1855 ± 19 |

Table 1b: Pb/Cs age results from parallel short core EN18232-2 (analysed by P. Appleby, pers. communication, 2019)

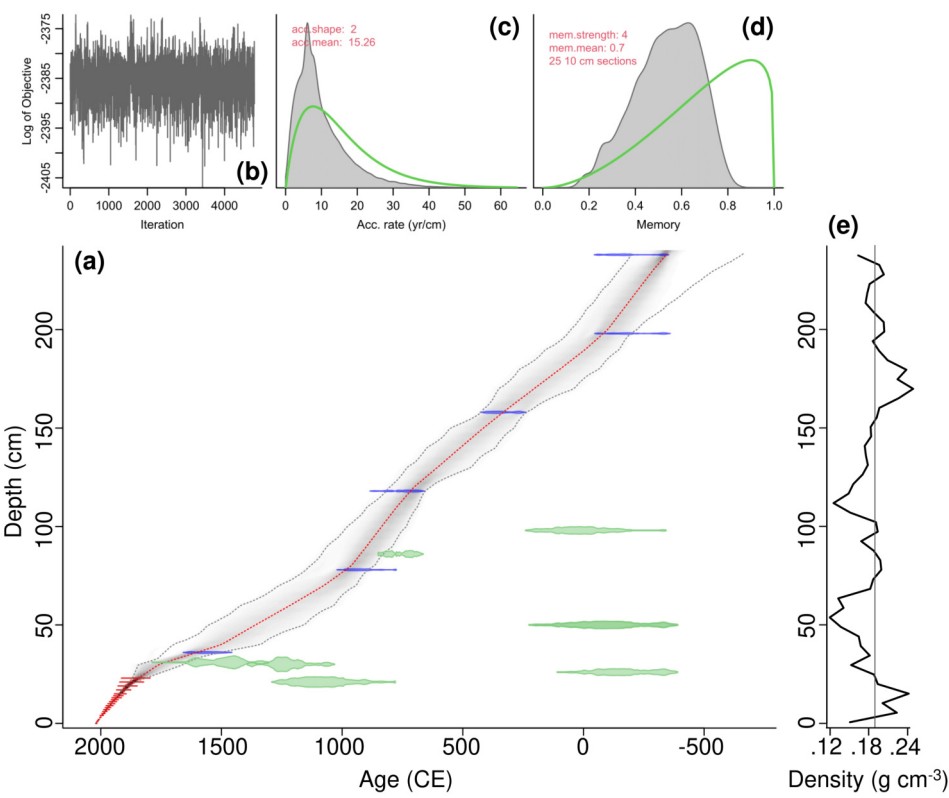

**Figure 2: (a) Bayesian age-depth model applying Bacon (Blaauw and Christen, 2011) to several types of determined sediment ages (red = Pb/Cs, blue = ¹⁴C bulk sediment, green = ¹⁴C macrofossils, grey lines = 2σ range, red line = median). (b) Model iteration log. (c), (d) Prior (green line) and posterior (grey area) distributions for accumulation rate and memory, respectively. (e) Dry bulk density of sediment core EN18232-3 (grey line = mean).**

## 3.2 Reconstructed fire regime

Samples of the charcoal record contain on average $8.1 \pm 5.1$ charcoal particles per cm³ of sediment (min: 0, max: 38.8). Interpreting the classic peak component as temporally restricted increases in fire activity, 50 such fire episodes within the continuously sampled core segment spanning the last c. 2200 yrs were identified. This results in a record-wide mean FRI of 43 yrs (min: 6, max: 594). A maximum estimate of this FRI, based on the reduced number of fire episodes by limiting one episode per peak with SNI >3, is 95 yrs (min: 12, max: 876). Results from the alternative local threshold variant slightly differ from the global threshold in expected ways (less fire episodes

identified during periods of high CHAR and more during periods of low CHAR), but do not produce majorly contrasting results (see Appendix A).

Classic and robust CHAR analysis distinguished four phases, representing different states of the fire regime (see Fig. 3). Phase 1 (c. 140 BCE to 600 CE) is characterised by relatively high CHAR of $0.83 \pm 0.43$ particles cm$^{-2}$
yr$^{-1}$ and numerous (n = 23) peaks (i.e. fire episodes, Fig. 3c), with a SNI that is slightly above 3 for the most part but slowly decreasing towards the following phase 2 (Fig. 3b). The mean FRI is 31 yrs (min: 6, max: 144). Robust CHAR shows a steadily increasing background component (Fig. 3d), whereas its peak component (Fig. 3e) remains at low levels. More fire episodes (n = 16) occur in the following, shorter phase 2 (600–900 CE), with a lower mean FRI of c. 14 yrs. It incorporates some of the highest peaks of the whole record using either the classic or robust
approach, resulting in a high CHAR of $1.3 \pm 0.54$ particles cm$^{-2}$ yr$^{-1}$ (including the maximum of 2.8 particles cm$^{-2}$ yr$^{-1}$) while SNI falls below 3. In the transition to phase 3 at around 900 CE a pronounced decrease in charcoal input leads up to a long period of few to no fire episodes (n = 6) and longer mean FRI of >60 yrs (min: 6, max: 594). Hence, CHAR of phase 3 (900–1750 CE) is low with $0.53 \pm 0.29$ particles cm$^{-2}$ yr$^{-1}$, while robust CHAR background remains below average. The SNI decreases to a record-wide minimum due to the lack of any CHAR
peaks above the global threshold in the second half of phase 3. Finally, phase 4 (1750–2018 CE) has higher CHAR ($0.61 \pm 0.47$ particles cm$^{-2}$ yr$^{-1}$) and more frequent occurrences of fire episodes (n = 5), with mean FRI sharply decreasing to 40 yrs (min: 6, max: 78). An outstanding peak around 1880 CE leads to a maximum SNI >6 during this phase. Although phase 4 sees increasing CHAR and fire frequency after the low CHAR of phase 3, charcoal input decreases to a minimum within the last century (mean CHAR of the last 100 yrs before core extraction =
$0.56 \pm 0.29$ particles cm$^{-2}$ yr$^{-1}$). Similarly, the robust CHAR sum and its components show increases within phase 4 (Fig. 3d, e), with two maxima in the robust peak component around the early 1800s and 1950 CE and a following decrease in CHAR (Fig. 3d-e). In general, the older half of the record (c. 140 BCE to 1000 CE) has higher mean CHAR and a higher variability ($0.97 \pm 0.5$ particles cm$^{-2}$ yr$^{-1}$) compared to the younger half (1000–2018 CE; CHAR = $0.51 \pm 0.32$ particles cm$^{-2}$ yr$^{-1}$). Even with added uncertainties from counting and the chronology, maxima
of robust CHAR mostly replicate periods of increased classic CHAR.

Over the entire record, 43.7% of charcoal particles belong to the smallest size class (150–300 μm), while 28.8% and 27.5% are part of the medium (300–500 μm) and large size classes (>500 μm), respectively. When assessed individually, more fire episodes are identified for smaller particles than for larger particles (Table 2). This is expected, as smaller particles tend to have a larger source area, potentially incorporating more fire events into the
signal (Conedera et al., 2009).

The most prevalent morphotypes present in the sediment are types F (elongated, 31.7%), M (irregular, 28.4%), S (angular/black, 20.6%), and B (angular/brownish-black, 7.2%), with all others (X, C, D, E, P) ranking at or below 3% each. Noticeably, two of the highest peaks of the record at c. 650 and 1880 CE are composed primarily of large (>500 μm), elongated (type F) particles. The total relative amount of type F particles seems to correlate with the largest particle size class, whereas type M is more closely associated with smaller particles (see Fig. 4 and Appendix B). Furthermore, the PCA indicates that there are rather weak grouping patterns of morphotype or size-class distributions in samples of increasing core depth and age, potentially reflecting a stable vegetation composition around the lake. The three charcoal morphotype groups show a similar temporal pattern for their background and peak component distributions (Fig. 6c, d), mostly mirroring the decreased variability in the second half of the record as described for the sum of all particles. However, when assessed individually, large particles have a generally lower variability than the other size classes, whereas the variability of irregular morphotypes is higher than that of elongated or angular particles (see Supplement).

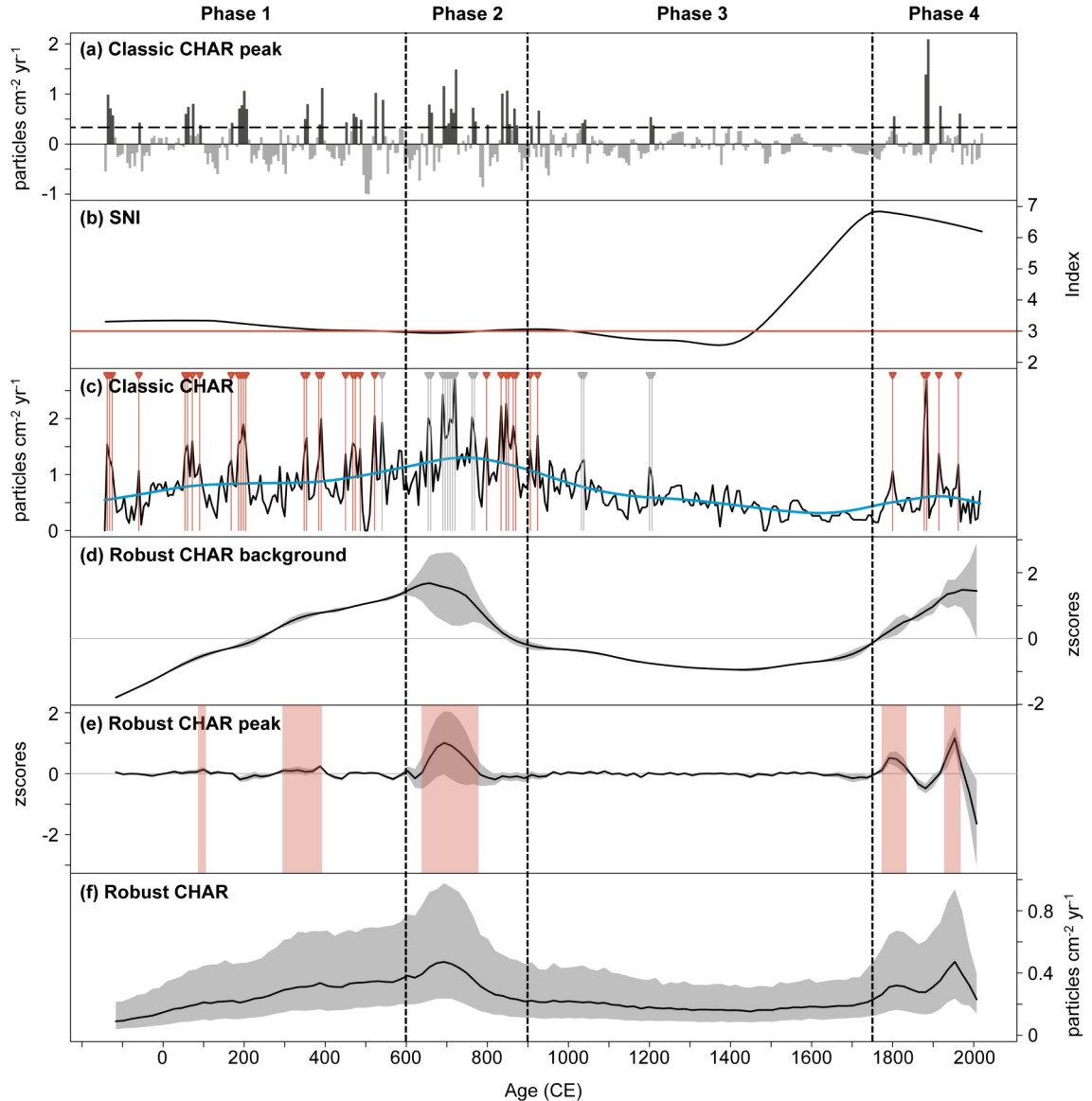

**Figure 3: Overview of the charcoal record using classic and robust analysis approaches. Vertical dashed lines mark the different phases of the fire regime. (a) Classic CHAR peak component (dark-grey bars = signal, light-grey bars = noise, dashed horizontal line = threshold). (b) Signal-to-noise index (SNI) of the classic CHAR peak component after Kelly et al. (2011) (red horizontal line = SNI cut-off value of 3). (c) Classic CHAR sum (black line = interpolated CHAR, blue line = LOESS representing the CHAR background component, red vertical lines = fire episodes with SNI >3, grey vertical lines = fire episodes with SNI <3). (d) Robust CHAR background component. (e) Robust CHAR peak component (red areas = above-average values). (f) Robust CHAR sum. For (d) – (f): black line = median, grey area = interquartile range.**

| Particle class (share of total) | Samples with non-zero values | Identified fire episodes | $FRI_{mean}$ | $FRI_{max}$ | $FRI_{min}$ | SNI >3 (%) |
|---|---|---|---|---|---|---|
| Elongated (36.2%) | 261 | 25 | 80.3 | 726 | 6 | 82.8 |
| Irregular (33.5%) | 261 | 80 | 27.3 | 228 | 6 | 92.8 |
| Angular (30.3%) | 260 | 28 | 78.7 | 456 | 6 | 78.4 |
| 150–300 µm (43.7%) | 287 | 57 | 37.5 | 372 | 6 | 65.1 |
| 300–500 µm (28.8%) | 282 | 45 | 48.5 | 444 | 6 | 56.2 |
| >500 µm (27.5%) | 269 | 18 | 118.9 | 954 | 6 | 80.3 |
| Total | 300 | 50 | 42.9 | 594 | 6 | 68.1 |

**Table 2: Reconstructed fire episodes, fire return interval (FRI) in years, and distributions of signal-to-noise index (SNI) >3 for charcoal morphotype and size classes, as well as the sum of charcoal particles over the last two millennia.**

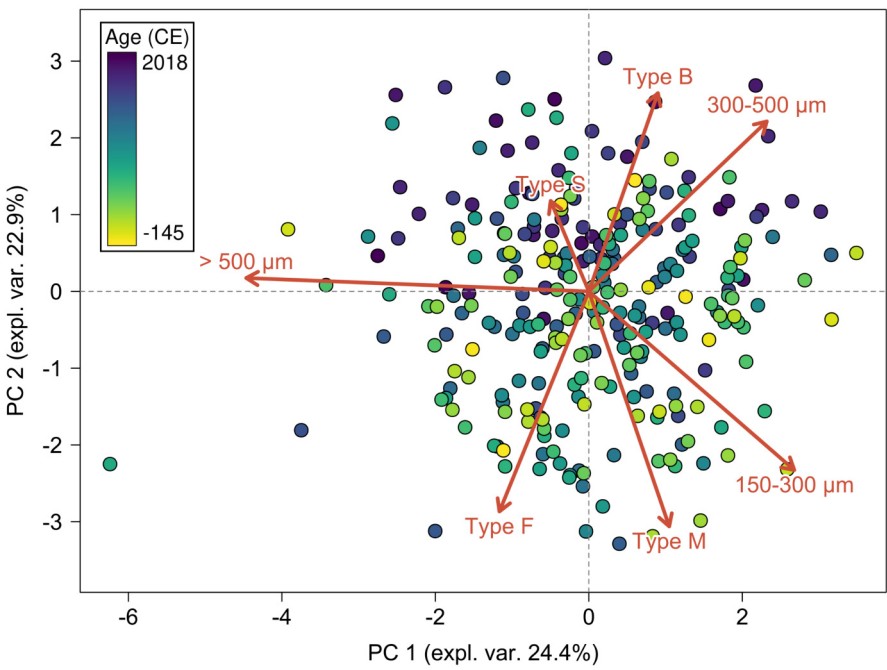


**Figure 4: Principal component analysis (PCA) of charcoal particles (size classes and individual morphotypes with a share of >5% of the charcoal record), with coloured dots representing potential grouping patterns of charcoal assemblages with increasing age.**

### 3.3 Vegetation history

The pollen and NPP record, covering the whole sediment core and reaching back c. 2350 yrs, generally indicates a relatively stable vegetation composition (Fig. 5). The dominant arboreal pollen (AP) types comprise most of the pollen spectra (average ratio of AP:NAP = 8.3:1) and include the trees and shrubs recorded around the lake. In descending order regarding their share of the pollen sum these are *Pinus*, *Betula*, *Picea*, *Abies*, *Alnus*, and *Larix*, with smaller amounts of *Salix*, *Juniperus*, and *Populus*. Non-arboreal pollen (NAP) types are predominantly
represented by Cyperaceae, followed by less abundant Poaceae, Ericales, and *Artemisia*. Despite similar general palynomorph distributions, pollen assemblages can be divided into two subzones, with the upper subzone (Ib) seeing intervals of increased variability in the shares of some tree pollen and Cyperaceae (around 10 and 120 cm depth, corresponding to c. 1950 and 700 CE, respectively). The lower subzone (Ia) demonstrates generally lower shares of *Abies* and Cyperaceae pollen. Charcoal particles and pollen types have mostly weak correlations without
statistical significance at $p > 0.05$, although they hint at weak associations between irregular morphotypes with AP being contrasted by those of angular morphotypes and NAP (see Appendix B).

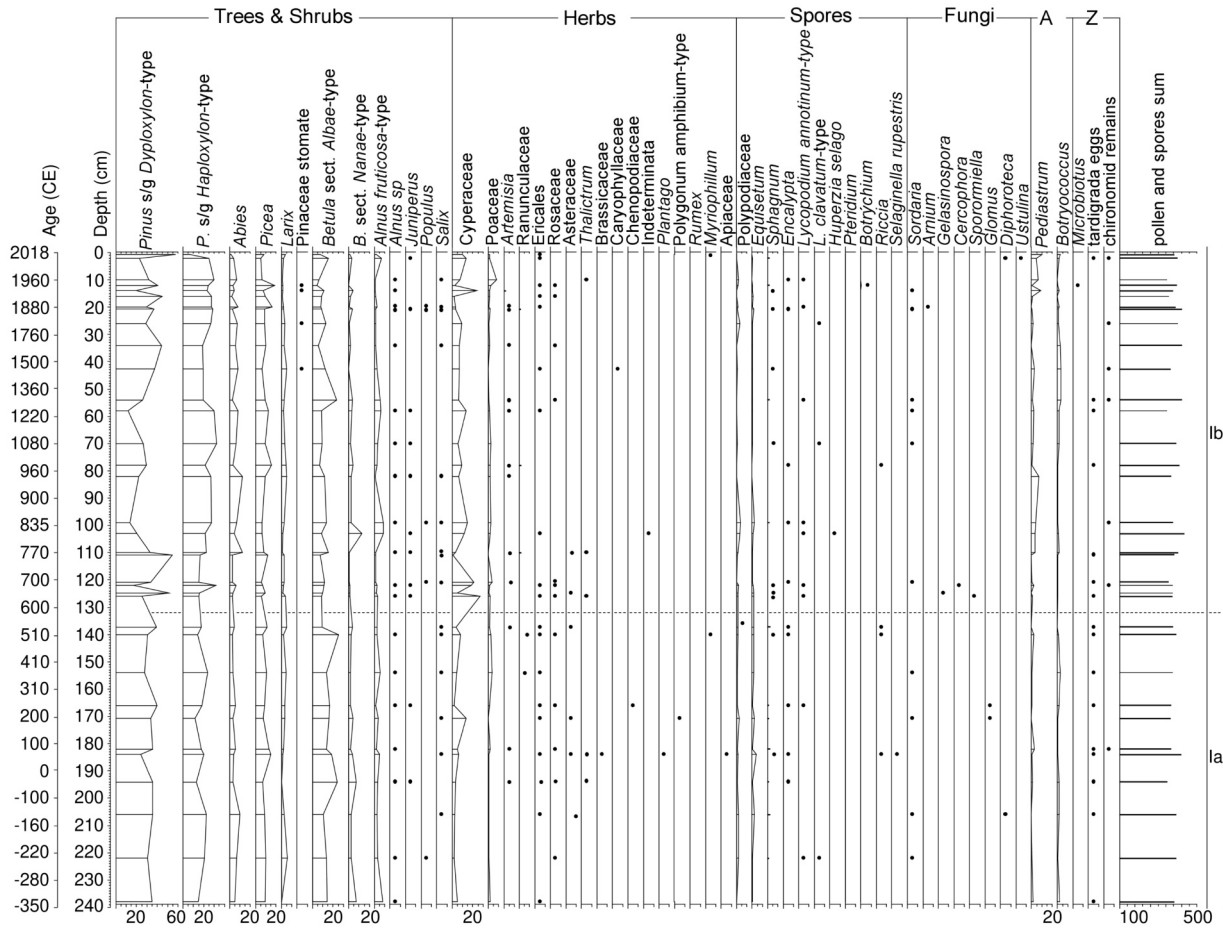

**Figure 5: Pollen and non-pollen palynomorph percentage diagram from sediment core EN18232-3 at Lake Khamra (dots represent pollen taxa <1%; A = algal remains; Z = invertebrate remains; horizontal dashed line = separation of subzones Ia and Ib).**

## 4 Discussion

### 4.1 Fire regime history of the last two millennia at Lake Khamra

We use the term "fire episode" instead of "fire event" when referring to identified peaks in the record. This is to highlight that multiple fires could have contributed to any peak in the charcoal record. Consequentially, the FRIs of this study should be regarded rather as "fire episode return intervals", marking the time span between periods of increased fire occurrence within the charcoal source area. This is because the relatively large water-surface area of Lake Khamra likely captures charcoal from a larger source area than smaller lakes. A larger source area of

charcoal is directly related to an increased number of fires that were able to contribute charcoal to the present record. The gentle and densely vegetated slopes framing the catchment limit secondary charcoal input (Whitlock and Larsen, 2001), thus emphasising a direct fire signal with predominantly primary input through the air. However, a higher number of captured fires from a larger source area also means that the comparability of FRIs reconstructed in this study to those obtained from smaller lakes or tree-ring chronologies may be limited, since those usually convey direct fire impact and are more locally constrained (Remy et al., 2018).

Although it has been shown that larger charcoal particles originate generally within a few hundred metres of a lake archive (Clark et al., 1998; Higuera et al., 2007; Ohlson and Tryterud, 2000), they have also been observed to travel further depending on vegetation and fire conditions (Peters and Higuera, 2007; Pisaric, 2002; Tinner et al., 2006; Woodward and Haines, 2020). As wildfires in the Siberian boreal forest are predominantly considered low-intensity surface fires (de Groot et al., 2013; Rogers et al., 2015), the potential of the resulting convection to transport large charcoal particles is probably limited compared to high-intensity crown fires. We therefore assume a charcoal source area between a few hundred metres directly around the lake for low-intensity fires (Conedera et al., 2009) and increasing distance of up to several kilometres for more intense fires producing stronger convection, resulting in a total source area estimate of up to c. 100 km². Even though some extreme fires may well surpass this estimate and, occasionally, small fires within might fail to contribute sufficient amounts of charcoal, identified fire episodes in the charcoal record should still be biased towards fires closer to the lake, especially when they consist of predominantly large charcoal particles (Conedera et al., 2009; Whitlock and Larsen, 2001). The uncertainty regarding any source area estimate highlights the need for further spatial calibration studies. Also, it remains unknown whether the charcoal record is dominated more by close-proximity low-intensity fires or by high-intensity fires from a larger distance. An estimation of fire intensity from charcoal particles (e.g. measuring charcoal reflectance, Hudspith et al., 2015; or the charcoal's oxygen to carbon ratio, Reza et al., 2020) could potentially clarify the respective contributions and thus help with better constraining the source area.

The macroscopic charcoal record at Lake Khamra (Fig. 3) reveals gradually increasing, but relatively stable fire activity from 140 BCE to 600 CE (phase 1). A period of high fire activity takes place between 600 and 900 CE, expressed as higher CHAR and shorter FRIs (phase 2). It then transitions into an almost 600-year period without any identified fire episodes and low CHAR (phase 3). From around 1750 CE the modern fire regime begins to take shape (phase 4), with regularly identified fire episodes marking increasing fire frequency. However, the most recent levels of CHAR are still lower than those of the maximum in phase 2 and reach a minimum in the 20th century, meaning that the amount of modern charcoal accumulation is not unprecedented within the last c. 2200 yrs. The mostly rather low SNI, which is below 3 for c. one third of the record, might result from the lower intensity

surface fire regime found around Lake Khamra. Such fires probably create less distinct peaks than the high

intensity crown fires of other regions, especially considering the large lake size.

Robust CHAR (Fig. 3d–f), incorporating uncertainties from the age-depth model and charcoal counting, necessarily loses the original charcoal record's short-term variability. It needs to be noted that any uncertainty potentially arising from the assumed constant rate of old carbon input to the lake, underlying the sediment core's chronology, is not included here, as any changes in magnitude of this reservoir-like effect are impossible to

quantify within the scope of this study. This issue is common in studies in permafrost regions that use $^{14}$C age dating (Biskaborn et al., 2012; Colman et al., 1996; Nazarova et al., 2013; Vyse et al., 2020). In such instances, applying Pb/Cs age dating adds valuable non-carbon-related estimates of sediment accumulation rates for the upper part of a sediment core (Whitlock and Larsen, 2001). One way of potentially quantifying old carbon age offsets throughout a sediment core might be to use cleaned charcoal particles, if available in sufficient number, and

evaluate their age difference to bulk $^{14}$C samples (similar to Gaglioti et al., 2014). Since charcoal is assumed to be delivered to the lake directly via the air, it might provide better estimates of the real sediment age, even when other plant macrofossils fail to do so.

The general trend of fire regime changes in SW Yakutia over the last c. 2200 yrs captured by the CHAR background component (Fig. 6c) and described above, shares many similarities with a charcoal record from the

evergreen forest in the Tomsk region (c. 1500 km west of Lake Khamra; Feurdean et al., 2020). There, one period of exceptionally high CHAR was observed around 700 CE and then again starting around 1700 CE towards the present, in parallel with Lake Khamra's fire history. The onset of increased biomass burning around 1700 CE was also reconstructed using aromatic acids from an ice core on the Severnaya Zemlya archipelago (c. 2200 km north-west of Lake Khamra), and it indicates a sudden decrease at the beginning of the 20th century (Grieman et al.,

2017). However, the same study finds another maximum of biomass burning around 1500 CE, in contrast to a clear period without fire episodes from c. 1250–1750 CE at Lake Khamra. Although comparisons across such long distances and between different climate zones, archives, and fire proxies are likely to show differing results, some of the described trends seem to be recorded at several sites worldwide. A global charcoal record synthesis for the last two millennia by Marlon et al. (2008) indicates decreasing biomass burning from c. 1 CE towards the industrial

era, where, after a maximum around 1850 CE, it decreases with the onset of the 20th century. A potential explanation for this similar trend during the most recent centuries, observed across many study sites from different regions of the world (e.g. Dietze et al., 2019), could be the onset of industrialisation with an accompanying change in land use and subsequent fire management. However, charcoal records from Siberia are underrepresented in such

synthesis studies (Marlon et al., 2016), which, together with a lack of comparable records closer to SW Yakutia, underline the issue of sparse data in the region.

Even with the constraints imposed by the large source area at this study site, the reconstructed record-wide mean FRI of 43 yrs, incorporating both the exceptionally short mean FRI of phase 2 (14 yrs) and the long mean FRI of phase 3 (60 yrs, excluding the c. 600-yr-long period without identified fires), lies within the range of the few comparable studies in boreal European Russia or western Siberia. Even if the record-wide mean FRI would strongly overestimate the number of included fire episodes, we would not expect the true value to exceed the maximum estimate of 95 yrs. Barhoumi et al. (2019) found the shortest FRIs of the Holocene ranging from 40–100 yrs between 1500 CE and present day in macroscopic charcoal records from the northern Ural region. A mean FRI of 45 yrs during recent centuries was inferred by Feurdean et al. (2020). Other studies using tree-stand ages and fire scars in tree-ring chronologies suggest mean FRIs of 80–90 yrs for mixed larch forests between the Yenisei and Tunguska rivers c. 1000 km north-west of Lake Khamra since c. 1800 CE, although the FRI of individual study sites could be as short as c. 50 yrs (Kharuk et al., 2008; Sofronov et al., 1998; Vaganov and Arbatskaya, 1996). A mean FRI of 52 yrs was reported for the northern Irkutsk region c. 300 km west of Lake Khamra in the 18th century (Wallenius et al., 2011) and 50–80 yrs for some sites in the north-eastern larch-dominated forests (Kharuk et al., 2011; Schepaschenko et al., 2008). In general, fire frequency tends to increase with decreasing latitude due to higher solar radiation, longer fire seasons, and higher flammability of dry biomass (Ivanova, 1996; Kharuk et al., 2016, 2011), which, in addition to the large source area, likely contributes to a relatively short mean FRI at Lake Khamra when compared to studies set further north. Furthermore, studies based on tree ring chronologies or fire scars are only able to detect fires where trees survived or deadwood within the range of existing chronologies remains, whereas sedimentary charcoal can potentially capture all types of fires.

Enache and Cumming (2007) explain how a large catchment to lake-area ratio might favour secondary deposition of compact/stable morphotypes, whereas fragile morphotypes are more prone to fragmentation during surface runoff and thus rather represent primary input through the air. However, the catchment to lake-area ratio at Lake Khamra (23:1), as well as the share of fragile charcoal particles (types F, M, and S alone make up >80%), are both comparably large. This might indicate that morphotype distribution within the record is not controlled by potential filtering effects of secondary charcoal transport, but rather by the type of biomass burning. This is also implied by the predominantly primary charcoal input through the air due to the densely vegetated surrounding slopes, and mirrors the stable vegetation composition seen in the pollen record.

Experimental charring studies have shown how different types of vegetation produce varied charcoal appearances (Jensen et al., 2007; Mustaphi and Pisaric, 2014; Pereboom et al., 2020). Pereboom et al. (2020) found elongated

charcoal particles after experimentally burning tundra graminoids, potentially hinting at the origin of the many elongated type F particles at Lake Khamra. However, these type F particles quite closely match the appearance of charred *Picea* needles reported by Mustaphi and Pisaric (2014). Together with the potential of more intense fires producing larger charcoal particles (Ward and Hardy, 1991), this could mean the two previously noted peaks of CHAR (c. 1880 CE at 19.5–20.5 cm depth and 650 CE at 124.5–125 cm; both dominated by high shares of type F

particles >500 µm) are evidence of higher-intensity fires burning conifer trees more severely and within a few kilometres of the lake shore. Charring experiments with local vegetation and a regionally adapted morphotype classification scheme would potentially benefit future studies by providing clear ground-truthing for links between morphotypes and vegetation.

## 4.2 Drivers of fire regime variations

### 4.2.1 Vegetation

The overall stable vegetation composition during the time covered by the charcoal record, as implied by the pollen and NPP record (Fig. 5), indicates that vegetation changes were unlikely to be the main driver behind changes in the fire regime, and/or that changes in fire regime did not lead to large-scale shifts in vegetation composition. Similarly, no prominent shift in charcoal morphotype composition, and hence in the type of biomass burned over

time, can be inferred (Fig. 4; Fig. 6c, d). Although some studies draw a similar conclusion (e.g. Carcaillet et al., 2001 in eastern Canada), this result contrasts with many other studies from the Eurasian and North American boreal zones, where vegetation changes were found to be closely connected to changes in fire regimes (Barhoumi et al., 2019, 2020; Feurdean et al., 2020; Gavin et al., 2007; Kelly et al., 2013). A reason for this might be that studies on longer timescales capture long-term concurrent trends in both fire and vegetation that are not observable

in a record that only captures the last c. 2200 yrs (e.g. glacial to interglacial changes in vegetation distribution and temperature; see Marlon et al., 2013). However, on a shorter, multi-decadal timescale, phases with more Cyperaceae pollen (sedges) in the Lake Khamra record and a higher ratio of evergreen to deciduous arboreal pollen types coincide with periods of high fire activity in phases 2 and 4 (see Fig. 6b). This could be due to either the ability of sedges to quickly settle on freshly disturbed and cleared out forest areas, and/or sedges growing in wetter

areas, possibly right at the lake shore, which are spared by fires (Angelstam and Kuuluvainen, 2004; Isaev et al., 2010; Ivanova et al., 2014). An increased proportion of evergreen trees might enable more intense crown fires. Indirectly, dry periods could lead to receding lake levels and thus an increase in both shoreland sedges and fire ignitions. However, such clear links are difficult to infer without hydrological data. In addition to differences in proxy source area and taphonomy between macroscopic charcoal and pollen grains, a variety of factors likely

obscures traces of potential fire impacts: surface fires in the deciduous forests in central Yakutia mostly result in the elimination of only a share of the affected tree population depending on fire intensity (Matveev and Usoltzev, 1996). This might not be enough to leave behind a clear mark in the pollen record, which also represents a source area that is probably way larger than the area affected by fire. Herbs or shrubs, on the other hand, may recover too quickly for changes to be detected in our record with a median temporal sampling resolution of 6 yrs. Any potential

mixing processes and the residence time of pollen grains before settling in the lake sediment may further diminish visibility of a fire impact (Campbell, 1999; Fægri et al., 1989). Furthermore, reconstructions of *Larix* dynamics based on fossil pollen are affected by the limited pollen dispersal distance of larch trees, as well as poor preservation of their pollen grains (Müller et al., 2010). This can lead to an underestimation of *Larix* in fossil pollen records (Edwards et al., 2000) and thus complicate the evaluation of larch tree dynamics. The relatively low

proportion of fossil *Larix* pollen at Lake Khamra, despite *Larix gmelinii* being a predominant tree taxon within the study area, may well reflect that issue. This indicates how future studies, aiming at specifically comparing past fire regime changes with *Larix* population dynamics, may benefit from including plant macrofossil analysis, if possible (e.g. Birks, 2003; Stähli et al., 2006). Due to a lack of macrofossils in the Lake Khamra sediment core and their suggested prolonged terrestrial residence time, as implied by the chronology, this was not an option in

the present study. These factors, together with a remaining ambiguity in morphotype classification, likely explain the rather weak correlation of the pollen and charcoal records.

**4.2.2 Climate**

Although it has been demonstrated that the timing and extent of supposedly ubiquitous warmer or cooler climatic phases are in fact heterogeneous (Guiot et al., 2010), evidence for their occurrence in Siberia is seen in several

proxy studies (Churakova Sidorova et al., 2020; Feurdean et al., 2019; Kharuk et al., 2010; Osborn and Briffa, 2006), albeit less pronounced when it comes to vegetation response in the West Siberian Lowland (Philben et al., 2014). Neukom et al. (2019) show how such climatic periods arising from averaged reconstructions at many individual study sites are not spatially or temporally coherent on the global scale and conclude that environmental reconstructions "should not be forced to fit into global narratives or epochs". This might be especially true for

studies of a single site, using chronologies that have [14]C reservoir effects. However, the low fire activity in the latter half of phase 3 (900–1750 CE) strikingly coincides with the Little Ice Age (LIA), when in many regions of the Northern Hemisphere a cooler climate prevailed from c. 1400 to 1700 CE. In contrast to this, high fire activity during phase 2 not matching the proposed timing of the warmer Medieval Climate Anomaly (MCA, c. 950 to 1250 CE) demonstrates the limitations of such comparisons based solely upon the [14]C-dated segment of the charcoal

record (estimates of LIA and MCA durations from Mann et al., 2009). Due to a lack of regional studies, the PAGES Arctic 2k temperature reconstruction (McKay and Kaufman, 2014) was used to provide a comparison between the reconstructed fire activity and large-scale changes in Arctic climate north of 60°N (Fig. 6a). This synthesis of circumpolar temperature reconstructions incorporates many records, although datasets from Siberia are sparse and thus it is underrepresented compared to Greenland, North America, and Europe. However, climate at Lake Khamra

is likely to be strongly influenced by the conditions further north, as Arctic temperatures affect the strength of the AO, which has been indirectly linked to fire activity further south (Balzter et al., 2005; Kim et al., 2020). The reconstructed PAGES Arctic 2k temperature provides evidence for a colder climate around c. 1600 CE, coinciding with the LIA and low fire activity at Lake Khamra. The following onset of increased fire frequency in phase 4 (c. 1750 CE onwards) is concurrent with a gradual increase in Arctic temperatures during the last two centuries (Fig.

6a, c), although not exceeding the maximum in fire frequency of phase 2. The older half of the charcoal record (before c. 1000 CE) does not match the temperature reconstruction as clearly. This might be due to a more regionally constrained climate, or a consequence of the lack of comparable data about humidity, which also affects fire activity (Brown and Giesecke, 2014; Power et al., 2008). A larch tree-ring based, c. 1500-yr-long reconstruction of summer vapour pressure deficit in north-eastern Yakutia (c. 2000 km from Lake Khamra)

indicates that the modern, increasing level of drought stress is not yet unprecedented, being surpassed by a high vapour pressure deficit during the MCA (Churakova Sidorova et al., 2020). This is similar to trends observed in the fire reconstruction at Lake Khamra. Another possibility for a less clear relationship between Arctic temperature and fire regime changes in the older half of the record is that the assumed constant impact of old carbon on the $^{14}$C age dating might be less pronounced in this older millennium. Yet, an impact of colder Arctic temperatures on the

reconstructed low fire activity in phase 2 at Lake Khamra seems probable. Since we can currently rely only on this one record, more palaeoclimatic reconstructions and high-resolution charcoal records from the region could greatly improve the validity of such inferred links between climate and the fire regime.

### 4.2.3 Human activity

The discrepancy of last century's low CHAR just after a phase of increasing fire frequency and rising Arctic

temperature could potentially be a sign of direct and indirect consequences of human activity around Lake Khamra. The anthropogenic influence on fire regimes may be the most difficult to quantify due to missing information about the kind and extent of human fire use and management throughout time, and the complex disentanglement of other drivers such as climate and vegetation (Marlon et al., 2013). Although Lake Khamra is located in a sparsely populated region, humans have historically been shaping the surrounding landscapes by building forest winter

tracks and roads (c. 0–30 km distance), logging (c. 10 km distance), and building villages and towns (c. 30 and 40 km distance, respectively). It has also been shown that even very remote lake systems in Yakutia have been affected by human activity from industrialisation (Biskaborn et al., 2021). Yakutia has been populated by humans since at least c. 28,000 BCE (Pitulko et al., 2004), although the population first noticeably increased at the end of the LGM around 17,000 BCE (Fiedel and Kuzmin, 2007), eventually forming the indigenous hunting and reindeer

herding tribes of Evens, Evenks, and Yukaghirs (Keyser et al., 2015; Pakendorf et al., 2006). Between 1100 and 1300 CE (in the first half of phase 3 in the charcoal record) the Sakha people moved in from the south, pressured by an expanding Mongol empire (Fedorova et al., 2013). They established a new and distinct form of semi-nomadic livelihood based on horse and cattle breeding (Pakendorf et al., 1999). Fire was mainly used at hearths to provide warmth and light, but also to sustain grasslands for grazing (Kisilyakov, 2009; Pyne, 1996). Population density

likely started to increase rapidly after Russian colonisation in the early 17th century (Crubézy et al., 2010), and even more drastically with the onset of new industries such as wide-scale logging and mining in the 20th century (Pyne, 1996). When compared to the pastoralist societies that existed up to that point, anthropogenic influence on the fire regime likely increased at the end of phase 3 (c. 1700 CE onwards) and throughout phase 4, after colonisation and industrialisation (Drobyshev et al., 2004). It has been shown that human livelihoods and the

mentality towards fire use can often better explain shifts in fire regimes than population density alone (Bowman et al., 2011; Dietze et al., 2018). For example, a formerly smaller population could have relied on practices like slash-and-burn agriculture until there was a transition towards more industrialised, urban livelihoods and a new focus on active fire suppression to protect forestry resources despite an increasing population (Dietze et al., 2019; Marlon et al., 2013). This is also thought to explain a pronounced decrease in boreal fire activity within the last

century in tree-ring studies from central Siberia (300 km west of Lake Khamra) and Fennoscandia (Wallenius, 2011; Wallenius et al., 2011). Forest roads and clearings could have acted as fire breaks, while the emergence of fire suppression in Russia was conceived as early as 1893 CE and later led to the founding of the first aerial firefighting unit in 1931 CE (Pyne, 1996). Adding to this, slash-and-burn agriculture was officially banned at the end of the 18th century, but was likely still practiced frequently up until the early 20th century (Drobyshev et al.,

2004; Konakov, 1999; Kozubov and Taskaev, 1999). Higher fire activity after c. 1750 CE, marking the onset of phase 4, therefore coincides with both a rapid increase in anthropogenic activity and land development, as well as a warming Arctic climate. Low CHAR within the last century on the other hand might be a consequence of a cultural shift towards seeing fire as a hazard to ban, control, and suppress. Whereas high fire frequency in the past corresponded with high amounts of biomass burned, the recent century also sees increasing fire frequency with

decreasing total biomass burned (Fig. 6c, d). This might indicate that the current fire regime differs from that

experienced by indigenous and Sakha people a few hundred years ago, now potentially consisting of more frequent, but smaller fires. A better understanding of fire use and management of the various Yakutian societies throughout history is needed to judge the extent of the human influence on fire regimes in relatively remote regions, especially since it has the potential to obscure the effects of recent global warming in fire reconstructions.


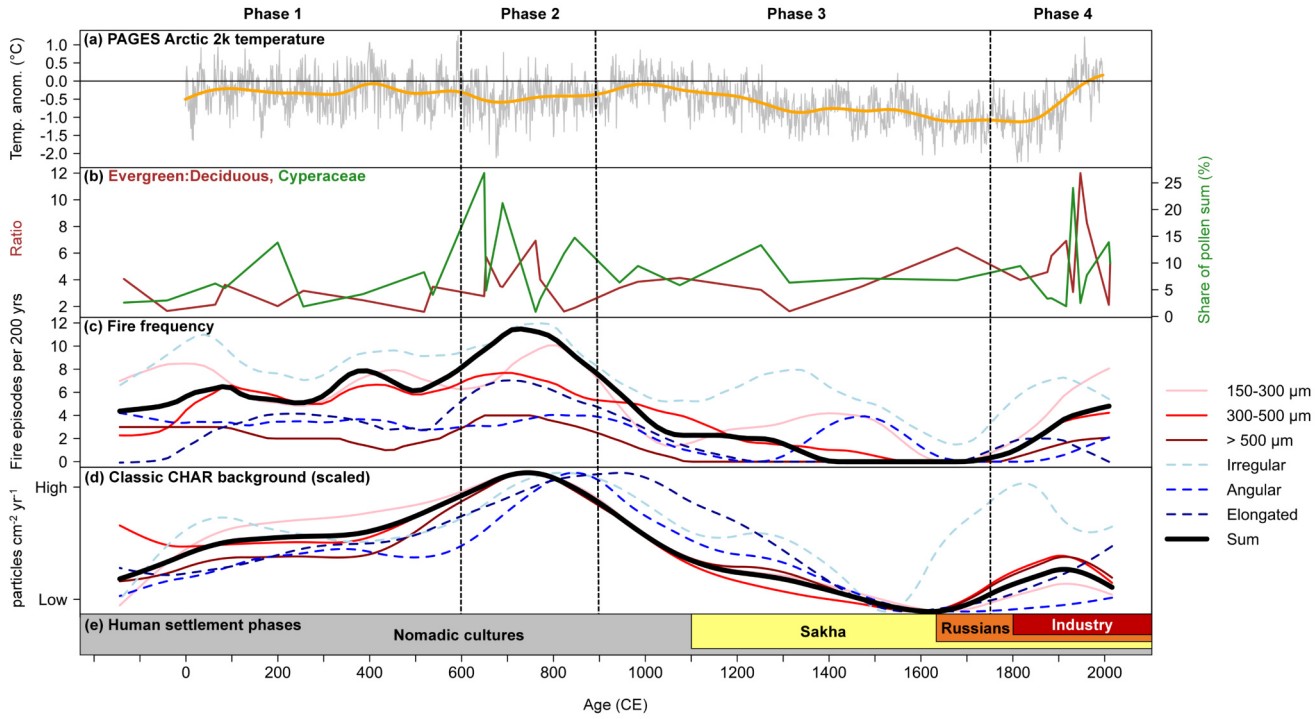

**Figure 6: Comparison of the charcoal record with climate, vegetation, and general human settlement phases. Vertical dashed lines mark the different phases of the fire regime. (a) PAGES Arctic 2k temperature (grey curve = original annual data, yellow line = LOESS). (b) Selected vegetation proxies. (c) Fire frequency; compiled, smoothed frequencies**
**of fire episodes for the sum of particles and individual size classes/morphotypes. (d) Compiled classic CHAR background components for the sum of particles and individual size classes/morphotypes, scaled to equal extents for comparison. (e) General human settlement phases, referenced in the text. For (c)-(d): Lines represent size and morphotype classes as well as the sum of charcoal particles.**

## 5 Conclusions

With its continuous sampling scheme and high median temporal resolution, the macroscopic charcoal record at Lake Khamra provides first insights into changes of the boreal fire regime of the last c. 2200 yrs in eastern Siberia, where comparable data are still lacking. Recent levels of charcoal accumulation at Lake Khamra are not

unprecedented within the last two millennia. The reconstructed fire regime changes do not coincide with large-scale shifts in vegetation composition, although short-term increases of evergreen trees and sedges broadly coincide with periods of increased biomass burning around 700 and 1850 CE (phases 2 and 4). Also, low fire activity from c. 900 to 1750 CE (phase 3), expressed as long FRIs and low charcoal accumulation, corresponds to a colder Arctic climate during the LIA. Despite the generally low population density, increased anthropogenic forcing after the colonisation of Yakutia by the Russians in the early 17th century might have contributed to an increase in fire frequency, together with rising temperatures. Although northern regions have been warming rapidly in recent decades, charcoal input to the lake has been minimal during the last century, coinciding with new fire management strategies and a ban on fire-related agricultural practices. The mean FRI of 43 yrs, with a maximum estimate of 95 yrs, is within the range of published literature for the wider region and incorporates a range of individual values of up to almost 600 yrs. The large lake size may be an important factor behind these generally shorter FRIs, since it is associated with a large charcoal source area of several kilometres from the lake and thus captures more fires compared to more locally constrained studies. Overall charcoal accumulation (classic CHAR background component) and the frequency of identified fire episodes seem to be directly related to each other for the majority of the record.

Although this new charcoal record improves data availability from eastern Siberia, more reconstructions, especially from distinctly deciduous regions, are needed to form a detailed analysis of past fire regimes in the Siberian boreal forest. An improved understanding of both fire activity and its drivers throughout history will eventually enable a meaningful assessment of the presence and future of Siberian wildfires and their consequences.

# Appendix

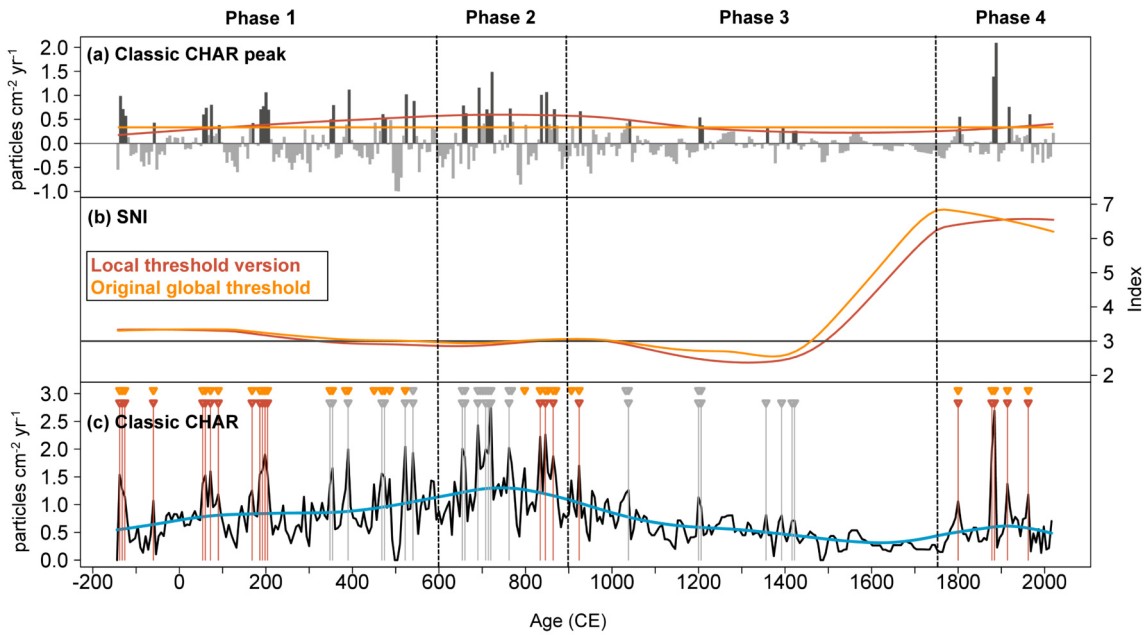

**Appendix A: Classic CHAR comparing the alternative local threshold (red) to the global threshold (orange). Vertical dashed lines mark the different phases of the fire regime. (a) Classic CHAR peak component (dark-grey bars = signal, light-grey bars = noise, coloured lines = threshold versions). (b) SNI after Kelly et al. (2011) (black horizontal line = SNI cut-off value of 3). (c) Classic CHAR sum (black line = interpolated CHAR, blue line = LOESS representing the CHAR background component, red/orange marks = fire episodes for local and global threshold versions, respectively, with colour = fire episodes with SNI >3, in grey = fire episodes with SNI <3).**

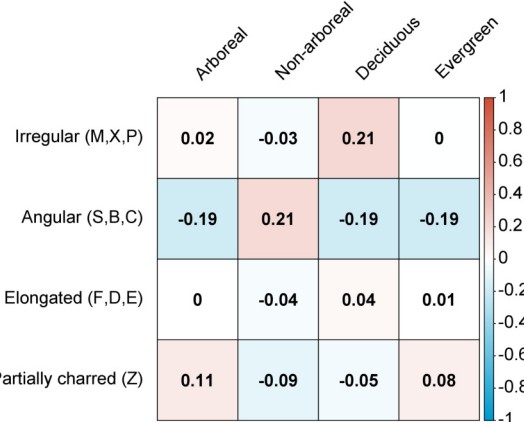

**Appendix B: Correlations (Kendall's τ) of charcoal morphotype classes with selected pollen groups.**

### Code and data availability

The R script used to analyse the charcoal record presented in this study can be accessed in the Zenodo database: http://doi.org/10.5281/zenodo.4943274 (Glückler and Dietze, 2021). The data presented in this study is available in the PANGAEA database (https://doi.org/10.1594/PANGAEA.923773; Glückler et al., 2020) and will be uploaded to the Global Paleofire Database (https://ipn.paleofire.org).

### Supplement

See separate Supplement file.

### Author contribution

ED and UH conceived and designed the study. LP, UH and SK organised the expedition to Yakutia. SV and BB collected the samples and conducted field measurements. RG and SV sampled the sediment core and performed the lab analysis. RG conducted all the charcoal proxy analysis, supported by ED and SK. AA conducted the pollen and non-pollen palynomorph analysis. BW and ED analysed remote-sensing data and created maps of the study location. RG wrote the paper with inputs from ED. All the authors reviewed the final manuscript.

### Competing interests

The authors declare that they have no conflict of interest.

### Acknowledgements

This study has been supported by the ERC consolidator grant Glacial Legacy of Ulrike Herzschuh (#772852). Ramesh Glückler was funded by AWI INSPIRES (INternational Science Program for Integrative Research in Earth Systems). Elisabeth Dietze was funded by the German Research Foundation, project DI 2544/1-1 (#419058007). Stuart Vyse was financially supported by the Earth Systems Knowledge Platform (ESKP) of the Helmholtz Foundation. We thank Nadine Bernhardt for her help working with the sediment core and Cathy Jenks for providing English proofreading, as well as the participants of the joint German-Russian expedition Yakutia 2018 for their support. We would further like to thank Philip Higuera, Angelica Feurdean and one anonymous

referee for providing valuable remarks for improvements to this paper. This study is a contribution to the PAGES-endorsed International Paleofire Network.

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
