# Peer review of "Wildfire history of the boreal forest of southwestern Yakutia (Siberia) over the last two millennia documented by a lakesedimentary charcoal record"

_Biogeosciences, 2020_

## Referee Comment (RC1) · Philip Higuera (Referee) · 16 Dec 2020

****General comments****

The paper presents well-developed datasets from what I imagine was a hard-earned lake-sediment record in a region lacking long-term fire history information. The text is well written. The graphics are clear and well-developed. The new "robust" charcoal analysis approach is refreshing. The community needs fire history information from this part of the boreal forest, and this is well motivated in the introduction.

[Figure]

[Figure]

Based on the comments below, this record does not seem well-suited for peak analysis and interpretation of peaks as individual fire events. Given lake size, a surface-fire component in the fire regime, and chronological uncertainty from old-carbon effects, interpreting total charcoal (concentration and/or accumulation rates) and a smoothed derivation may be more justified. The spatial integration of this record, given the large lake size, could be an advantage to help more reasonably compare general trends in charcoal accumulation (as a proxy for regional biomass burning) to regional climate, vegetation, human history.

My two main concerns are described below:

[1] Chronology: I appreciate the many challenges of developing chronologies from boreal lakes, and the authors are upfront about these challenges. Nonetheless, some important limitations of the chronology remain and seem to not be transferred through to the interpretation of the proxies. Most concerning is the assumption that a single old carbon offset applied to the entire core. The same approach was used in Vyse et al. (2020), but without further citation or justification. How robust is this assumption; does it also assume the rate of permafrost thaw is non-varying over time? Any additional information supporting these assumptions would help potentially quell these concerns.

In line 370 in the current paper, the authors note "…any changes in the magnitude of this reservoir-like effect are impossible to quantify." But, couldn't that assumption be tested by dating the charcoal that is assumed to be deposited at the same time as the sediment? The macrofossil dates likely reflect materials with a long terrestrial residence, but the charcoal pieces – to be interpreted as they are – should reflect relatively instantaneous deposition. A similar approach (based on non-charred terrestrial macrofossils) has been used to quantify variability in the age offset over time in a tundra lake: Gaglioti et al. 2014 (https://agupubs.onlinelibrary.wiley.com/doi/pdf/10.1002/2014JG002688).

The chronology issue is important given that (i) CHAR calculations are a function of

sediment accumulation rates, and (ii) the record is interpreted at fairly fine temporal scales - e.g., Phase 2 is only 300 yr long, and there are interpretations of the LIA and MCA. Interpreting fire history at these scales is already pushing the limits of 14C-based chronologies, and the added uncertainty of dating bulk sediment with known old carbon contributions seems additionally constraining.

[2] Charcoal peak analysis and interpretation:

(i) Lake size: The rationale and tools developed for peak analysis (e.g., decomposition approach in general, and as reflected in CharAnalsyis) assume a small lake surface area, and that charcoal primarily comes from airborne deposition. For example, most lakes used for peak analysis are < 10 ha (e.g., Alaskan lakes summarized by Hoecker et al. 2020). A lake with 4.6 km2 (460 ha) surface area is quite different, and this distinction is key to point out and carry though the interpretation of the record. The large lake size could be an advantage – integrating more area than a small lake – but it does not lend itself then to interpreting individual peaks in the record.

For example, interpreting intervals between peaks is significantly different for a lake this size vs. a lake < 10 ha, since the large lake integrates a much larger area. At a minimum, it's confusing to compare mean FRIs from a lake with such a large surface area to mean FRI estimates from tree rings (summarized over a small area), small lakes, or modern fire history records (e.g. summarized as fire rotation periods).

(ii) Peak analysis and consecutive samples above a threshold: The peak analysis presented here appears to consider all samples above the threshold – even in adjacent samples – as peaks and thus fire events. This is quite different from "classic" CharAnalsyis, as implied in the methods, and this is unlike examples I am familiar with from the literature (e.g., CharAnalysis or predecessors CHAPS and Charster). Typically, it is recognized that a single event can create a charcoal peaks that span multiple samples (and the first or maximum value is used as the peak date); other approaches would benefit from explicitly describing the framework and rationale used, and provide any

empirical support. The result are challenging to accept: e.g. adjacent samples above the threshold are interpreted as distinct fire events, such that a mean FRI of 14 yr is inferred for Phase 2 (line 304). Is there any modern calibration work that supports this type of interpretation (i.e., that consecutive samples above a threshold indeed reflect different fires)?

(iii) Peak analysis in a surface-fire regime: More broadly, a surface-fire regime is not expected to create distinct peaks in CHAR (as noted in the text). Peak analysis is generally considered most suited for high-severity fire regimes. Thus, it's not surprising that the SNI is at or below 3 for nearly $\frac{1}{2}$ of the record; the large lake size likely also contributes to the low SNI values. Interpreting peaks in CHAR from a low-severity fire regime, with a record with SNI $\approx< 3$, should recognize that many low-intensity surface fires are likely missed. But again...all of this in in the context of small lakes – the larger lake adds more "noise" to this rationale, and calls into question the value/meaning of return intervals in the first place.

****Specific comments****

L 44: Consider Kelly et al. 2016 (Nature Geoscience 6:79-82) as a useful reference for boreal forest carbon balance changing with changing fire regimes.

L 165: Nice way to save sediment here, with the dual pollen-charcoal subsampling.

L 189-190: Nice way to help account for some counting uncertainty.

L 206: This is slightly misleading, as there appears to be important differences between what was done here and what is implemented in CharAnalayis. For example: (i) CharAnalsyis does not identify adjacent samples above the threshold as peaks, as is done here; and (ii) it appears that the Gaussian mixture model used here may be different from the one used in CharAnalysis, if not the actual algorithm, then in the way it's applied. See notes on Fig. 1, below. Overall, it would be more accurate to say something like: "First, we used a set of analyses to decompose the charcoal records

into peak and background signals, similar to well-established approaches applied in CharAnalysis." Upon reading the original text...it really sounds like the same methods of CharAnalysis were translated into R (which I wish were true!).

L 116: A better paper to describe how a Gaussian mixture model is used to identify a threshold would be Gavin et al. 2006 (Ecology 87:1722-1732 – first to use Gaussian mixture model) or Higuera et al. 2011. To my knowledge this method was not established yet in 2003.

L 218: This trade-off between "Longer window widths..." that yield a higher SNI values and "a strong averaging of the record" is in part what motivates the use of local thresholds (e.g., in CharAnalysis). Local thresholds also reduced the impacts from any changes in CHAR due to change in sediment accumulation rate.

L 221: This rationale justifying why peaks are interpreted when the SNI is consistently < 3 is not very convincing.

L 225: Unlike in Dietze et al. (2019), the differences between this "robust" approach and the "classic" approach applied is more challenging to make sense of in this record. E.g., the "robust CHAR peak" in panel (e) is hard to reconcile with "classic" results, particularly in Phase 2.

L 301, 204: Are these mean FRIs of 31 and 14 years because multiple peaks in a row are interpreted as fire events? I keep double checking this...but this must be the case. I don't understand how consecutive samples above the threshold are interpreted as separate/independent fire events. This needs some empirical (and/or theoretical) support.

L 334 – Figure 3: The threshold identified (Fig. 1a) seems very low – e.g., there are many negative samples (anomalies) that would exceed the threshold, were it inverted to be negative, particularly in Phase 1-2. Conceptually, samples exceeding the same threshold value below 0 suggest something is off in the parameters, as there is no

interpretation for negative departures beyond the threshold. This type of record, even though short, is the type that motivates local thresholds, as there are changes in both the background and variability in CHAR over the different phases. *But again. . .this is usually in the context of smaller lakes.

L 344: Why not plot this based on age, instead of depth? All other analyses are presented by age – it seems odd to have this plotted by depth.

L 368-371: As noted above, this seems like a major constraint of the chronology, and thus interpretation. It's good that it's pointed out here, but it's then hard to reconcile this with interpreting changes with the LIA or phases that are 300 yr long.

L 385: Would we expect one site to necessarily reflect regional or global patterns in fire activity, at these smaller time scales? If so, it would be worth including the potential mechanisms for such synchronous fire activity.

L 405: Yes – shorter intervals between peaks in small charcoal, compared to peak in large charcoal – make sense based on a larger source area for smaller charcoal. It's key to tell readers what spatial scale, approximately, you think this record integrates, prior to this point in the text. The spatial scale reflected is key to interpreting the FRI values described above.

L 407-410: This comparison conflates a bit the difference between "just dispersal" vs, "enough charcoal to create a peak that is distinct from background charcoal." Large pieces can travel far. . .and distinct peaks can still be strongly biased towards "local" fires.

L 412-414: This assumption of the spatial scale reflected by the charcoal records would be much more useful if it came before presenting the FRI information – the meaning of FRI (and mean FRI) is contingent upon the spatial scale reflected or integrated across. Given the circumference of the lake, what does this translate to in terms of km^2? That's the key piece of information.

L 441: Could this "contrast" between the current study and others, to some extent, reflect the differences in temporal scale? The current study is "only" 2000 yr long, whereas several of the studies cited span much longer time periods. Mechanisms for vegetation change vary over these different scales.

L 443-444: And...the very large spatial footprint integrated by the pollen record in this lake is key here. A clear pollen signal would require persistent vegetation change over a large area.

L 467-468: Doesn't this also directly apply to comparing the fire history record reconstructed here (i.e., one site, with a chronology subject to 14C reservoir effects) to any proxy with a well-constrained chronology?

L 543: Some citations would help identify the other studies noted here.

---

## Referee Comment (RC2) · Angelica Feurdean (Referee) · 17 Dec 2020

The manuscript by Glückler et al., is timely executed study of past wildfire dynamics and associated drivers in Eastern Siberia. The manuscript is based on high quality data and statistical analysis, is clearly written and well referenced. The study concluded that, at this temporal scale, it is climate and human impact, rather than vegetation driving the past millennial to centennial changes in wildfire activity.

One of my main concern is the approach to the vegetation. The study is conducted

in Larix (larch) dominated forests of Siberia. Larix is notoriously underestimated in the pollen records, and this should make it difficult to accurately reconstruct its past dynamics. Plant macrofossil analysis, mostly abundant needles (deciduous conifer tree), could help constrain its past dynamics, however, this is appropriately done in small basins or cores close to the lake margin, which is not the case of this site. The pollen record presented in this study neither show that Larix was an abundant taxon nor that its proportion have changed in the past, which likely highlights the problems above. I suggest adding a few lines acknowledging the problem of reconstructed past Larix dynamics based on pollen.

The other concerns on the chronology and charcoal peak analysis have been highlighted by the other reviewer (P. Higuera).

Specific comments:

l. 63-63 Barhoumi et al.2019 study lies in European Russia not in Siberia, please correct

l.100 Note here that Larix gmelinii as one of the main tree taxa

l.177 What do you mean by this? You broke the charr particles with a needle?

Palynological analysis l. 191-194, Ok, but it would be useful to also state the resolution of pollen samples l. 200 Which subsequent analysis? How were the % calculated?

l. 240 Please state what exactly was desired with the correlation between charcoal and pollen (not vegetation), and which pollen types were chosen and why. I got the feeling that the results from pollen record are minimized.

L.320 What was the propose on running CHAR on separate grain size and morphotypes? This is not stated in the methods.

l.325 Is there a difference between angular S and B morphotypes?

l.334 Do you mean similar pattern for all charcoal morphotypes?

3.3 Vegetation history. I suggest adding the pollen diagram into the main paper. Would it make sense / increase visibility, to use continuous lines i.e, curves instead of bars to show trends in the pollen record? The past trends in vegetation are described in just two lines 353-355, an expansion of this is needed. In the pollen diagram (A1) there are two zones and at minimum the composition /differences between the 2 should be highlighted.

l. 347-355 I am confused by this statement, why is now Larix listed last? Additionally, the tree pollen composition may reflect that of the surrounding forest, but not the proportion. For ex Larix is one of the dominant taxa in the forest presently (according to the introduction and study area), however it was only found with scarred pollen grains.

l. 393 Barhoumi et al.2019 not in west Siberia

l. 405 Agree but this needs to be stated earlier in the methods and results.

l. 414 A few hundred meters is really little.

l. 419 How then?

l. 425. I am a bit confused here. To which letter /type do the elongated type belong? I believe that burning graminoids would produce elongated charr particles also in Siberia, judging from other studies on the L:W ratio. However, there are others fuel types that have elongated morphologies.

4.2.1 Vegetation l. 435. Apart of the problem of large site and footprint on pollen record, please think whether biases with Larix pollen could have also 'falsely' contributed to this monotony. Are there other pollen diagrams in the region to document how vegetation composition varied regionally?

l. 448 increased proportion of . . .

l. 450 do you imply that pollen is more problematic than charcoal?

l.478 which? Please state the age range?

l.     480   you   may   want   to   look   at   the   moisture   record
https://doi.org/10.1016/j.quascirev.2019.105948

l. 550 given the large lake size, could charcoal input over time have been affected by different locations of fire in the catchment and the subsequent charcoal delivery into the lake?

---

## Referee Comment (RC3) · Anonymous Referee #3 · 21 Dec 2020

This is a carefully prepared and well written manuscript. The topic is certainly important: fire records are needed from Siberia. Despite the detailed analyses, this is nevertheless a difficult site to interpret. I am not convinced by the results or their current interpretation for the following main reasons, which have been articulated in detail by the other reviewers.

1) The chronology is very difficult. The offset of 1000 yr in the bulk sediment series may be approximately right, given there is carbonate bedrock in the vicinity. However, the mixed-up macrofossil dates suggest that material of different ages becomes incor-

porated into the sediment matrix, so why not the charcoal?

2) The FRI's seem extraordinarily short. An average of 43, in a ∼2000 yr series that has a quiescent period of ∼600 yr is high, and when broken down into zones/phases, estimated FRI levels of 14 yr do not sound at all realistic. Nothing as short as this is reported from the region.

3) The relevance of calculating FRI's for different types of charcoal morphology is not explained and no convincing implications of doing this, or the results, are presented.

Thus despite excellent detailed methodology, it would be difficult to draw much that is useful from this study. This study would be better presented as simpler types of time series and together with the pollen (as suggested by the other reviewers); it is far preferable to treat the data appropriately than to develop complex analyses that could well provide a misleading picture of events.

Other comments

L80 useful to mention any estimates of FRI here in description of region, seeing this is a fire study.

L187 need to clarify a bit more about the samples that had no char analysis; was it just the 11?

L206 well established

L250 Between lines 250 and 263 there is repetition, this part needs re-writing. In general, the discussion of the radiocarbon dates is long and over-complicated. It would help the reader to place a statement about how mixed the radiocarbon dates are right up front and state the whole problem much more directly.

L266 sentence beginning "(II) Macrofossil 14C ages. . .." Not quite sure what this means

L345 The relevance of the PCA is hard to see; more explanation in caption would help. Given the lack of impact of the morphology data (and Fig 4 only mentioned in results

once), this could be omitted.

L400. "In general, FRIs increase with latitude due to lower incoming solar radiation, shorter fire seasons, and lower flammability of moist biomass (Kharuk,2016; Kharuk et al., 2011), which likely contributes to a relatively short mean FRI at Lake Khamra. " Explain further? This site has a short FRI therefore............it is further south than other sites? The argument is not clear.

L 419 argument is a bit hard to follow in sentence beginning "However, the present "

L459 there is low peak frequency during much of what might be thought of as the MO and higher one toward the end of the LIA, so there is not a very good fit to climate – this is over-interpreted, especially given the caveats provided and the difficult chronology

---

## Short Comment (SC1) · 22 Dec 2020

I wish to submit a short comment on one part of the paper regarding the robust charcoal method originally described in Dietz et al. 2019 in PlosOne.

Incorporating uncertainties into proxy records, including both the age uncertainty and the uncertainty of the proxy itself, is important especially when comparing periods within a core and when comparing sites. It is overdue to include uncertainty in the analysis of individual sediment charcoal records. So, it is great to see this extension of

the methods from the 'ensemble' approach from Blarquez et al.

My comment addresses the resampling methods used for estimating the uncertainty of the sediment accumulation rates. The robust method uses the age estimate of each sample (described as a mean and sd, but it could also be a PDF from an age-depth model), and selects ages from that PDF. Ages are generated independently for all samples, and only ages in adjacent samples that are in chronological order are retained. This results in some very slow sedimentation rates. This is acknowledged in the 2019 paper: "A comparison showed that robust fluxes were smoothed, but underestimated absolute mean fluxes due to strongly overlapping pdfage of adjacent samples at 1 cm sample resolution. Hence, we averaged the raw proxy and age values of three adjacent samples before robust flux calculation."

I am not sure how the averaging as described makes the CHAR influx values more comparable to the original influx values—did you average three samples in non-overlapping segments, thus increasing the age difference of adjacent composite samples? The presented robust CHAR values are small compared to the raw data. I do not see this averaging step in the supplied code.

When the PDFs of adjacent samples are overlapping (<2 sd), the median age difference of the simulated ages is greater than the difference in the mean ages of the best-fit age-depth relationship. This is demonstrated in the attached figure. I think such small age differences occur in the majority of Holocene sediment records. The net effect is that as a core varies in sedimentation rate, the simulated sedimentation rate will have an increasing effect from the overlapping PDFs as the sedimentation rate decreases. This results in different effects of the analysis occurring in different parts of the same core. Variability in simulated sedimentation rates will not vary directly with the variation in the sedimentation of the best-fit age-depth model.

An alternative approach to simulating sedimentation rates: use the output from bacon or clam, which saves many simulated runs of age-depth relationships. These can be

used directly in the robust char calculations. The advantage here is that the simulated age-depth relationships preserves the monotonic age-depth pattern. The necessary ages for using this approach are in objects saved by the bacon and clam programs. (objects called info, dat, or chron). Clam and bacon can apply age uncertainties to proxy records directly. However, you have more flexibility by using the set of simulated age-depth relationships.

Code for the attached figure: ad <- matrix(NA,nrow=101,ncol=2) k<-1 for(i in seq(200,300,1)){ age.diffx <- (rnorm(10000,mean=i,sd=20)-rnorm(10000,mean=200,sd=20)) ad[k,1]<-i ad[k,2]<-median(age.diffx[age.diffx>0]) k<-k+1 }

plot(ad[,1]-200,ad[,2],xlab="difference in best estimate of sample ages (yr)",ylab="median of difference of simulated ages (yr)",xlim=c(0,100),ylim=c(0,100)) abline(a=0,b=1) text(x=10,y=90,"sd of each sample=20 yrs",pos=4)
* * *
[Figure]

**Fig. 1.**

---

## Editor Comment (EC1) · Sandy Harrison (Editor) · 26 Dec 2020

The reviewers have provided some very helpful comments on the methodology underpinning this study, which I hope that the authors will address in some detail in their response. Specifically, please address the following issues:

1) the reliability of the chronology and the impact of chronological uncertainties on the conclusions 2) the appropriateness of the calculated fire return intervals 3) the resampling methodology and its impact on the results 4) how/why the fire peak analysis

differs from conventional methods and the impact of this on the reliability of the re-constructions 5) the vegetation data used and particularly the degree to which under-representation of key pollen types is likely to have impacted your conclusions.

Sandy Harrison (handling associate editor)

---

## Author Comment (AC1) · 19 Feb 2021

Dear Philip Higuera,

thank you for your review of our manuscript and the time and effort put into it! We welcome your constructive feedback, which we value and without doubt improves this manuscript. Please find below your original comments in black and our author responses in green:

****General comments****

The paper presents well-developed datasets from what I imagine was a hard-earned lake-sediment record in a region lacking long-term fire history information. The text is well written. The graphics are clear and well-developed. The new "robust" charcoal analysis approach is refreshing. The community needs fire history information from this part of the boreal forest, and this is well motivated in the introduction.

Thank you, we appreciate your assessment!

Based on the comments below, this record does not seem well-suited for peak analysis and interpretation of peaks as individual fire events. Given lake size, a surface-fire component in the fire regime, and chronological uncertainty from old-carbon effects, interpreting total charcoal (concentration and/or accumulation rates) and a smoothed derivation may be more justified. The spatial integration of this record, given the large lake size, could be an advantage to help more reasonably compare general trends in charcoal accumulation (as a proxy for regional biomass burning) to regional climate, vegetation, human history.

We mostly agree with this general statement, which is why we chose to use such smoothed derivations of the charcoal record in the comparisons to climate/vegetation/human history (background component, which results from a LOESS applied to charcoal influx, and smoothed "peak frequency"). However, your comment made clear to us that some of our data description and its interpretation do not match the broader level of detail as provided by the lake archive and its chronology in its current state. We will therefore revise passages with the goal of capturing all factors that differentiate this record from smaller lakes or strictly local and well-constrained proxies such as tree ring studies. Please refer to our responses on your individual remarks below for more detail on the applied revisions. An important suggested addition will be a new paragraph at the beginning of the discussion describing clearly how our terms (e.g., "fire episode", "FRI") are defined, what signals they capture based on our archive and the lake's size, and which uncertainties need to be stressed when interpreting the data (see our response to [2] (i)).

My two main concerns are described below:

[1]     Chronology:

I appreciate the many challenges of developing chronologies from boreal lakes, and the authors are upfront about these challenges. Nonetheless, some important limitations of the chronology remain and seem to not be transferred through to the interpretation of the proxies. Most concerning is the assumption that a single old carbon offset applied to the entire core. The same approach was used in Vyse et al. (2020), but without further citation or justification. How robust is this assumption; does it also assume the rate of permafrost thaw is non-varying over time? Any additional information supporting these assumptions would help potentially quell these concerns. In line 370 in the current paper, the authors note "...any changes in the magnitude of this reservoir-like effect are impossible to quantify." But, couldn't that assumption be tested by dating the charcoal that is assumed to be

deposited at the same time as the sediment? The macrofossil dates likely reflect materials with a long terrestrial residence, but the charcoal pieces – to be interpreted as they are – should reflect relatively instantaneous deposition. A similar approach (based on non-charred terrestrial macrofossils) has been used to quantify variability in the age offset over time in a tundra lake: Gaglioti et al. 2014 (https://agupubs.onlinelibrary.wiley.com/doi/pdf/10.1002/2014JG002688).

Regarding the assumption of a stable old carbon effect over time:

As an integral part of a paleoenvironmental study that needs to derive accumulation rates from an age-depth model, we understand that the assumption of a constant old carbon related age-offset raises concern. While at the sediment surface we can estimate the magnitude of the offset, we do not have this possibility below c. 25 cm depth. Unfortunately, as you stated, the dated plant macrofossils, as a potential way to quantify bulk sediment offsets through time, seem to be part of the problem at this lake, as they might derive also from older permanently frozen deposits that have thawed.

If we would ignore the (important) $^{210}$Pb/$^{137}$Cs ages, a possible and often used approach would be to only adjust the surface $^{14}$C age to the year of core extraction and then connect that to the other, non-adjusted $^{14}$C ages (e.g. Biskaborn et al., 2012, 2013). This approach of assuming a recent $^{14}$C surface age is also used in studies that do not have any $^{14}$C surface age estimates (e.g. Katamura et al., 2009; Klemm et al., 2016). Lacourse and Gajewski (2020) found that about two thirds of 80 recently published age-depth models manually assign a surface age without reporting the details of that decision, thus may not be able to see any potential carbon offset issues. In this case, however, such an approach not only assumes that any input of old carbon only happened during recent times (which we know from deeper, mixed macrofossil ages that it did not), but it would also create a strong and unlikely shift in sedimentation rate between the top two $^{14}$C ages. The Lake Khamra sediment appearance does not offer any evidence for such a change, its homogenous composition rather being a reason for suspecting a linear sedimentation rate. This is underlined by reports of stable lake conditions and thermokarst processes in central Yakutia during the late Holocene (at least up to the rapid warming during past decades, e.g. Ulrich et al., 2019; Pestryakova et al., 2012).

In summary, we recognize that assuming a constant age offset due to old carbon input throughout the record has its limitations. However, the evidence from $^{210}$Pb/$^{137}$Cs, bulk and macrofossil $^{14}$C dates and sediment appearance neither hint at a highly varying age offset, nor at a highly variable sedimentation rate. That being said, we fully agree that we have to carry this underlying and known uncertainties through to a proper interpretation of our results, together with the robust CHAR analysis that explicitly considers known age and analytical uncertainties. We will additionally follow recommendations on best practices for age-depth model reporting by Lacourse and Gajewski (2020), by adding the lab numbers to the table of $^{14}$C dated samples and clearly stating that MICADAS uses AMS $^{14}$C dating.

Regarding age dating of charcoal particles:

Dating charcoal particles, which are suspected to derive mostly from primary input, is a very reasonable idea! However, after careful consideration, we think that within this study it is not feasible for the following reasons: (i) scarce sediment material – the sediment core provided valuable material from a helicopter-based expedition in Yakutia, where liners of 6 cm diameter had been used. Even with the combined charcoal and pollen extraction, most of recovered sediment now has either already been used or is required by other analyses from colleagues, making it difficult to obtain new and potentially larger sediment samples for the purpose of extracting higher amounts of charcoal for age dating. (ii) limited amount of charcoal in the existing samples – even samples with the highest number of charcoal particles are suspected to barely provide enough material for a reliable dating process on their own, seeing how 6 out of our 15 macrofossil samples were also too

small (<= 10 µg C). We would thus have to combine/destroy multiple charcoal samples across the record, potentially also losing the precision needed to effectively constrain the age offset. Unfortunately, together with the rigorous preparation steps for radiocarbon dating of very small charcoal particles as recommended by Bird (2013), these factors rule out a test of this method within the scope of this present study. Despite this, we do think that it would definitely be worth a try as we now increasingly start to use 9 cm liners, yielding more sediment material to work with, which is why we will include that option in future studies and refer to the inspiring study by Gaglioti et al. (2014) you kindly provided.

The chronology issue is important given that (i) CHAR calculations are a function of sediment accumulation rates, and (ii) the record is interpreted at fairly fine temporal scales - e.g., Phase 2 is only 300 yr long, and there are interpretations of the LIA and MCA. Interpreting fire history at these scales is already pushing the limits of 14C-based chronologies, and the added uncertainty of dating bulk sediment with known old carbon contributions seems additionally constraining.

The assumption of a stable age offset, the main concern voiced about the chronology above, would only change the absolute timing of all samples deeper than the second $^{14}$C age, but less affect the distribution of CHAR values relative to each other. This is because with or without the offset, we have no reason to assume abrupt and strong shifts in the sedimentation rate based on the sedimentology. For the upper samples, on the other hand, we provide a well-constrained $^{210}$Pb/$^{137}$Cs chronology and do have evidence for the age offset, including an estimate of its magnitude. The four phases are therefore not dependent on the age information.
However, we agree that any comparison to temporally constrained events (like the LIA) must be adequately justified. We will revise the discussion of climate forcing (4.2.2) to be clearer about the uncertainties in any such comparison, e.g. by stating: "[…] high fire activity during phase 2 not matching the proposed timing of the warmer Medieval Climate Anomaly […] demonstrates the limitations of such comparisons based solely upon the $^{14}$C-dated segment of the charcoal record […]." However, other studies on similar timescales (e.g. Churakova Sidorova et al., 2020; Feurdean et al., 2019) have reported that such climatic phases left behind a visible impact on various proxy reconstructions in Siberia. With a marked decrease in fire activity probably being climate-related, it seems likely that low CHAR in phase 3 could indeed correspond to a colder climate as reported for the LIA. We think this assumption is justified also as the reliable, near-surface $^{210}$Pb/$^{137}$Cs ages allow to constrain the LIA time frame better than older climatic periods.

 [2]      Charcoal peak analysis and interpretation:

(i)      Lake size: The rationale and tools developed for peak analysis (e.g., decomposition approach in general, and as reflected in CharAnalsyis) assume a small lake surface area, and that charcoal primarily comes from airborne deposition. For example, most lakes used for peak analysis are < 10 ha (e.g., Alaskan lakes summarized by Hoecker et al. 2020). A lake with 4.6 km2 (460 ha) surface area is quite different, and this distinction is key to point out and carry though the interpretation of the record. The large lake size could be an advantage – integrating more area than a small lake – but it does not lend itself then to interpreting individual peaks in the record.

For example, interpreting intervals between peaks is significantly different for a lake this size vs. a lake < 10 ha, since the large lake integrates a much larger area. At a minimum, it's confusing to compare mean FRIs from a lake with such a large surface area to mean FRI estimates from tree rings (summarized over a small area), small lakes, or modern fire history records (e.g. summarized as fire rotation periods).

You are absolutely right, we did not pay enough attention to a discussion of the influence of lake size and the subsequent meaning of our descriptive terminology. Hence, we have now revised the

discussion part of the manuscript and include the definitions of terms used, as well as summarize the archive's benefits and downsides at the beginning, before comparisons to any other studies are made. Now, charcoal peaks and their meaning as fire episodes are clearly put into context of the lake size, the surface fire regime, and an estimate of charcoal source area, which in turn enables to reader to see how the FRIs of this study may differ from others. The newly added paragraph reads as follows:

"We use the term "fire episode" instead of "fire event" when referring to identified peaks in the record. This should highlight that multiple fires could have contributed to any peak in the charcoal record. Consequentially, the FRIs of this study should be regarded rather as "fire episode return intervals", marking the time span between periods of increased fire occurrence within the charcoal source area. This is because the relatively large water surface area of Lake Khamra likely captures charcoal from a larger source area when compared to smaller lakes. A larger source area of charcoal is directly related with an increased number of fires that were able to contribute charcoal to the present record. The gentle and densely vegetated slopes framing the catchment limit secondary charcoal input (Whitlock and Larsen, 2001), thus emphasizing a direct fire signal with predominantly primary input through the air. However, a higher number of captured fires from a larger source area also means that the comparability of FRIs reconstructed in this study to those obtained from smaller lakes or tree ring chronologies may be limited, since those usually convey direct fire impact and are more locally constrained (Remy et al., 2018).
Although it has been shown that larger charcoal particles originate generally within a few hundred metres of a lake archive (Clark et al., 1998; Higuera et al., 2007; Ohlson and Tryterud, 2000), they have also been observed to travel further depending on vegetation and fire conditions (Peters and Higuera, 2007; Pisaric, 2002; Tinner et al., 2006; Woodward and Haines, 2020). As wildfires in the Siberian boreal forest are predominantly considered low-intensity surface fires (de Groot et al., 2013, Rogers et al., 2015), the potential of the resulting convection to transport large charcoal particles is probably limited compared to high-intensity crown fires. We therefore assume a charcoal source area between few hundred metres directly around the lake for low-intensity fires (Conedera et al., 2009) and increasing distance of up to several kilometres for more intense fires producing stronger convection, resulting in a total source area estimate of up to c. 100 km². Even though some extreme fires may well surpass this estimate and, occasionally, small fires within might fail to contribute sufficient amounts of charcoal, identified fire episodes in the charcoal record should still be biased towards fires closer to the lake, especially when they consist of predominantly large charcoal particles (Conedera et al., 2009; Whitlock and Larsen, 2001). The uncertainty regarding any source area estimate highlights the need for further spatial calibration studies. Also, it remains unknown whether the charcoal record might be dominated more by close-proximity low-intensity fires, or by high-intensity fires from a larger distance. An estimation of fire intensity from charcoal particles (e.g. measuring charcoal reflectance, Hudspith et al., 2015; or the charcoal's oxygen to carbon ratio, Sumon Reza et al., 2020) could potentially clarify the respective contributions and thus help with better constraining the source area."

Among some other minor changes regarding a clearer communication of our results, we will also rephrase the misleading statement on FRIs as "fire occurrences": "Interpreting the classic peak component as temporally restricted increases in fire activity, 50 such fire episodes […] were identified." Also, we added to the conclusion: "The large lake size may be an important factor behind these shorter FRIs, since it is associated with a large charcoal source area of multiple kilometres from the lake and thus capturing more fires compared to more locally constrained studies."

(ii)     Peak analysis and consecutive samples above a threshold: The peak analysis presented here appears to consider all samples above the threshold – even in adjacent samples – as peaks and thus

fire events. This is quite different from "classic" CharAnalysis, as implied in the methods, and this is unlike examples I am familiar with from the literature (e.g., CharAnalysis or predecessors CHAPS and Charster). Typically, it is recognized that a single event can create a charcoal peak that spans multiple samples (and the first or maximum value is used as the peak date); other approaches would benefit from explicitly describing the framework and rationale used, and provide any empirical support. The results are challenging to accept: e.g. adjacent samples above the threshold are interpreted as distinct fire events, such that a mean FRI of 14 yr is inferred for Phase 2 (line 304). Is there any modern calibration work that supports this type of interpretation (i.e., that consecutive samples above a threshold indeed reflect different fires)?

You are correct, our approach here considers all peaks reaching above the threshold as "fire episodes". We do not assume that one peak necessarily equals one individual fire, as it integrates multiple years over a fairly large source area. We rather think of fire episodes as periods of increased fire activity, based on one or more fires. From remote sensing observations we know that fires burned within Lake Khamra's catchment in 2006/7 and 2014 (Fig. 1b of the manuscript), just 8 years apart. Considering the median sample resolution of 6 years, such fires could well have been responsible for two consecutive peaks in the charcoal record. After fires, we expect the dense surface vegetation of the lake catchment to quickly recover (i.e., < 6 years), and together with the sedge belt and wetland around the lake acting as an efficient barrier for secondary input via surface runoff; both aspects have been observed in the field (see expedition reports from this region in Kruse et al., 2019; Fuchs et al., 2021). Additionally, any remaining charcoal that is deposited after a fire would have to spread across the large lake basin, probably leading to a smoothing rather than a second peak. This leaves charcoal fixation and sediment mixing processes as a remaining option for the distribution of one fire's charcoal across multiple samples, however, the charcoal record shows many individual peaks that are clearly distinct from their adjacent samples. This indicates that charcoal redistribution effects during sedimentation did not exceed the sampling resolution. Based on this rationale we decided to include every peak above threshold. To make this clear to the reader and provide this reasoning for any future discussion, we will include this reasoning briefly in the methods section. However, we will also feature a minimum estimate of fire episodes (and thus a maximum estimate of FRI) by considering only one of directly adjacent fire episodes that simultaneously have a SNI >3.

The methods paragraph will be revised as follows: "All peak component values exceeding the threshold were subsequently identified as signal (representing fire episodes) and marked when they overlapped with periods of SNI >3, indicating whether they are clearly distinct from surrounding noise. While usually in instances of multiple consecutive samples above threshold only the highest peak is recognized, we included all of them as fire episodes in this case. Recent fires within the lake's catchment that were just 8 years apart, as well as a predominantly primary input of charcoal and quick recovery of the filtering wetland vegetation around the large lake, lead to suspect that the ability of a single fire to create high charcoal counts in multiple samples is limited at this site. However, an absolute minimum estimate of the number of fire episodes was obtained by considering only one fire episode from a given peak in CHAR, and only those that are clearly separated from noise (i.e. they also overlapped with phases of SNI >3)."

Consequently, a maximum estimate of a mean FRI will be mentioned in the results and discussion sections next to the standard estimate of mean FRI at 43 yrs: "Even if this mean FRI would strongly overestimate the number of included fire episodes, we would not expect the true mean FRI to exceed the maximum estimate of 95 yrs."

 (iii)      Peak analysis in a surface-fire regime: More broadly, a surface-fire regime is not expected to create distinct peaks in CHAR (as noted in the text). Peak analysis is generally considered most suited for high-severity fire regimes. Thus, it's not surprising that the SNI is at or below 3 for nearly 1/2 of

the record; the large lake size likely also contributes to the low SNI values. Interpreting peaks in CHAR from a low-severity fire regime, with a record with SNI < 3, should recognize that many low-intensity surface fires are likely missed. But again...all of this in in the context of small lakes – the larger lake adds more "noise" to this rationale, and calls into question the value/meaning of return intervals in the first place.

We will include this important relationship between fire proximity and fire intensity in our revised discussion of charcoal source area: "Even though some extreme fires may well surpass this estimate and, occasionally, small fires within might fail to contribute sufficient amounts of charcoal, identified fire episodes in the charcoal record should still be biased towards fires closer to the lake, especially when they consist of predominantly large charcoal particles (Conedera et al., 2009; Whitlock and Larsen, 2001)."

So, even though the fire that burned approximately half of Lake Khamra's 107 km² catchment and beyond in 2014 was likely of lower intensity than usual forest fires elsewhere (MODIS-based mean fire radiative power of c. 76 MW, compared to generally higher values in Canada after Rogers et al., 2015), we would expect it to leave a distinct mark in the sediment archive. Unfortunately, the relationship between proximity and intensity is difficult to further assess for the fire episodes identified in the charcoal record due to the temporal mismatch of recent satellite observations and the sediment archive's coverage, as well as the difficulty of differentiating low- from high-intensity reconstructed fires in the sedimentary record so far. However, we agree that the predominant fire regime is indeed an additional reason for increased noise, and thus the generally low SNI, and as such we will include it in the updated manuscript in the discussion: "The mostly rather low SNI, which is below 3 for c. one third of the record, might result from the lower intensity surface fire regime found around Lake Khamra. Such fires probably create less distinct peaks than the high intensity crown fires of other regions, especially considering the large lake size."

****Specific comments****

L 44: Consider Kelly et al. 2016 (Nature Geoscience 6:79-82) as a useful reference for boreal forest carbon balance changing with changing fire regimes.

This is a great study to include here. We will add it as reference to L44 in the revised manuscript.

L 165: Nice way to save sediment here, with the dual pollen-charcoal subsampling.

Thank you! Since sediment material from expeditions is quite valuable, maybe this might benefit some other research by potentially enabling more proxies to be analyzed from the same sediment core.

L 189-190: Nice way to help account for some counting uncertainty.

Thank you, we will try to carry that through to future studies and see where we can improve on capturing this uncertainty.

L 206: This is slightly misleading, as there appears to be important differences between what was done here and what is implemented in CharAnalayis. For example:

(i) CharAnalsyis does not identify adjacent samples above the threshold as peaks, as is done here; and (ii) it appears that the Gaussian mixture model used here may be different from the one used in CharAnalysis, if not the actual algorithm, then in the way it's applied. See notes on Fig. 1, below. Overall, it would be more accurate to say something like: "First, we used a set of analyses to decompose the charcoal records into peak and background signals, similar to well-established

approaches applied in CharAnalysis." Upon reading the original text...it really sounds like the same methods of CharAnalysis were translated into R (which I wish were true!).

You are correct, this is indeed misleading. We are, of course, clarifying this part in the revised manuscript. Similar to your suggestion, L206 will read: "First, we applied a set of analyses referred to as "classic CHAR" (R script by Dietze et al., 2019), similar to the charcoal record decomposition in the well-established "CharAnalysis" (Higuera et al., 2009)." Furthermore, the repository and R script names associated with this manuscript will be changed to "CharcoalFireReconstruction".

L 116: A better paper to describe how a Gaussian mixture model is used to identify a threshold would be Gavin et al. 2006 (Ecology 87: 1722-1732 – first to use Gaussian mixture model) or Higuera et al. 2011. To my knowledge this method was not established yet in 2003.

You are right, Whitlock and Anderson (2003) describe the purpose of the peak component's threshold and not the specific method, so we will instead add "(Gavin et al., 2006; Higuera et al., 2011)".

L 218: This trade-off between "Longer window widths..." that yield a higher SNI values and "a strong averaging of the record" is in part what motivates the use of local thresholds (e.g., in CharAnalysis). Local thresholds also reduced the impacts from any changes in CHAR due to change in sediment accumulation rate.

We justified the use of a global threshold with the relatively uniform sedimentation rate as implied by the age-depth-model (with or without adjustment of the age offset), as well as the very homogenous vegetation composition that is thought to provide steady fuel types during the comparably short time covered by the record.
Nevertheless, we tested whether the application of a tentative local threshold would result in improved SNI or a different interpretation of reconstructed fire history (see Fig. 1 below). To be comparable to the global threshold used in the manuscript, we calculated it in a similar manner, using the same window width that was applied to the background LOESS. Briefly, the Gaussian mixture model was applied to the positive peaks of the peak component in a moving window. The local threshold was then obtained by applying a default LOESS to the resulting values to minimize the impact of outliers. Expectedly, this local threshold version identified less fire episodes during phase 2, and more during phase 3. However, the difference is not so large as to change key points of our discussion. Also, this threshold variant does not seem to improve the SNI. Of course, other methods of smoothing and calculating a local threshold will likely yield slightly different results as well. With the improved definitions on the terms used in our discussion and more appropriate comparisons to other studies mainly based on the general distribution of peak and background CHAR, we view our interpretation as not being majorly affected by the choice of threshold method. For that reason, we suggest keeping the current and approach of a global threshold in the revised manuscript, but adding Fig. 1 below to the Appendix for the reader to compare. The local threshold test will be briefly described in 2.4 Statistical methods. The R script now also contains the option to apply this tentative local threshold (see https://github.com/rglueckler/CharcoalFireReconstructionR/tree/revised).

L 221: This rationale justifying why peaks are interpreted when the SNI is consistently < 3 is not very convincing.

We will remove the sentence starting in L221 in the updated manuscript. Instead, to more clearly describe our rationale for application of the SNI, we will add to L218: "[…] and marked when they overlapped with periods of SNI >3, indicating whether they are clearly distinct from surrounding noise." The SNI will furthermore now be used to provide a minimum estimate of fire episodes, as noted above.

L 225: Unlike in Dietze et al. (2019), the differences between this "robust" approach and the "classic" approach applied is more challenging to make sense of in this record. E.g., the "robust CHAR peak" in panel (e) is hard to reconcile with "classic" results, particularly in Phase 2.

We applied the robust approach to account for the age and counting uncertainties, inherent to any reconstruction, with the resulting trends being a conservative estimate of the changes in past fire regimes. The relatively large age uncertainties, as well as a high counting uncertainty of 20% when compared to Dietze et al. (2019), lead to a high degree of smoothing from resampled distributions. Therefore, we would not expect robust CHAR to mirror the peaks found in classic CHAR, but rather only those phases that are prominent enough to stand out even with these added uncertainties. However, we have revised the script behind the calculation of robust CHAR to, among some other minor changes, optionally include a sample aggregation step used in Dietze et al. (2019). This has so far not been implemented in this manuscript, and allows for a combination of multiple samples to better scale our high-resolution record to the large added uncertainties. With this revised script (available at https://github.com/rglueckler/CharcoalFireReconstructionR/tree/revised), we re-calculated robust CHAR and will include this new version in the updated manuscript and the diagrams in its Supplement. It does not change the general trends observed in the current version, but it improves the fit between classic and robust CHAR, especially in phase 4 (see comparison in Fig. 2 below).

L 301, 204:  Are these mean FRIs of 31 and 14 years because multiple peaks in a row are interpreted as fire events? I keep double checking this...but this must be the case. I don't understand how consecutive samples above the threshold are interpreted as separate/independent fire events. This needs some empirical (and/or theoretical) support.

Yes, consecutive fire episodes are part of the FRIs. The definition of fire episodes is important here, which we have not sufficiently explained in the current manuscript, as well as our rationale for including adjacent identified peaks in the FRIs. As we have laid out above, the revised manuscript provides descriptions of both aspects, while also featuring a maximum FRI estimate, based on the reduced number of fire episodes when adjacent ones from the same CHAR peak distribution are ignored (and only considering those with SNI >3).

L 334 – Figure 3: The threshold identified (Fig. 1a) seems very low – e.g., there are many negative samples (anomalies) that would exceed the threshold, were it inverted to be negative, particularly in Phase 1-2. Conceptually, samples exceeding the same threshold value below 0 suggest something is off in the parameters, as there is no interpretation for negative departures beyond the threshold. This type of record, even though short, is the type that motivates local thresholds, as there are changes in both the background and variability in CHAR over the different phases. *But again...this is usually in the context of smaller lakes.

Negative samples of the peak component visualization (Fig. 3a in the manuscript) are residuals from the subtraction of the background component from the CHAR timeseries. This means that those samples did not contribute any, or only some few, charcoal particles to the record. For this reason, the threshold is only based on the positive peaks within the peak component, as only they contain the information about fire episodes we are looking for. Negative samples that might exceed an inverted threshold are more frequent in phases 1 and 2 because of the higher variability in CHAR in these phases, resulting in more samples with few charcoal particles in phases of a higher background component. We tested whether capturing this difference in background component and variability in our peak detection made an important difference by testing a tentative local threshold as laid out for a similar remark above (also see results in Fig. 1 below).

L 344: Why not plot this based on age, instead of depth? All other analyses are presented by age – it seems odd to have this plotted by depth.

We agree, samples within the PCA will be plotted according to their age instead of depth in the revised manuscript.

L 368-371: As noted above, this seems like a major constraint of the chronology, and thus interpretation. It's good that it's pointed out here, but it's then hard to reconcile this with interpreting changes with the LIA or phases that are 300 yr long.

As described above, the phases are not set regarding chronological information or some compared climate data, but rather depend on the charcoal distribution in the sediment core. However, we understand that we can, among other improvements, better communicate the general chronological uncertainty in the discussion. Especially in 4.2.2, we will emphasize the limitation of comparing our record to climatic phases, e.g. by stating: "[…] high fire activity during phase 2 not matching the proposed timing of the warmer Medieval Climate Anomaly […] demonstrates the limitations of such comparisons based solely upon the $^{14}$C-dated segment of the charcoal record […]".

L 385: Would we expect one site to necessarily reflect regional or global patterns in fire activity, at these smaller time scales? If so, it would be worth including the potential mechanisms for such synchronous fire activity.

With a multitude of factors influencing fire activity on a regional level when considering shorter timescales, we would not necessarily expect that. Nevertheless, it seems that an increase in fires in the 18$^{th}$ to 19$^{th}$ centuries, followed by a decrease within the 20$^{th}$ century, is recorded at multiple sites across studied regions (examples are listed as references in this part of the manuscript). Since there are so many other differing factors between these various study sites, synchronizing mechanisms could include wide-spread climatic events (such as the LIA) that might have influenced fires regardless of the ecosystem they occurred in (for example, a likely longer snow season during the LIA could have reduced the length of the fire season). With the timeframe from 18$^{th}$ century to present days, population growth and cultural transformations during the industrialization could be a further candidate for such a mechanism, as described in Marlon et al. (2008) for a similar timescale to our record. Since this is a better reference to include here when compared to Marlon et al. (2013), which looks at the whole Holocene, we will update L387: "A global charcoal record synthesis for the last two millennia by Marlon et al. (2008) indicates decreasing biomass burning from c. 0 CE towards the industrial era, where, after a maximum around 1850 CE, it decreases with the onset of the 20th century. A potential explanation for this similar trend during the most recent centuries, observed across many study sites set in different regions of the world, could be the onset of industrialization with an accompanying change in land use and subsequent fire management. However, charcoal records from Siberia are underrepresented […]".

L 405: Yes – shorter intervals between peaks in small charcoal, compared to peak in large charcoal – make sense based on a larger source area for smaller charcoal. It's key to tell readers what spatial scale, approximately, you think this record integrates, prior to this point in the text. The spatial scale reflected is key to interpreting the FRI values described above.

We totally agree, and will include this information directly in a new paragraph at the beginning of the discussion as described above: "We therefore assume a charcoal source area between few hundred metres directly around the lake for low-intensity fires (Conedera et al., 2009) and increasing distance of up to several kilometres for more intense fires producing stronger convection, resulting in a total source area estimate of up to c. 100 km². Even though some extreme fires may well surpass this estimate and, occasionally, small fires within might fail to contribute sufficient amounts of charcoal,

identified fire episodes in the charcoal record should still be biased towards fires closer to the lake, especially when they consist of predominantly large charcoal particles (Conedera et al., 2009; Whitlock and Larsen, 2001).".

L 407-410: This comparison conflates a bit the difference between "just dispersal" vs, "enough charcoal to create a peak that is distinct from background charcoal." Large pieces can travel far...and distinct peaks can still be strongly biased towards "local" fires.

We agree, it is the distinct peaks that are commonly interpreted to represent local fires, but not necessarily the individual particles a peak is made of. We illustrated the different size classes' travel distance to constrain a rough estimate of source area. This part will be rephrased and featured in the source area discussion as noted above: "Although it has been shown that larger charcoal particles originate generally within a few hundred metres of a lake archive (Clark et al., 1998; Higuera et al., 2007; Ohlson and Tryterud, 2000), they have also been observed to travel further depending on vegetation and fire conditions (Peters and Higuera, 2007; Pisaric, 2002; Tinner et al., 2006; Woodward and Haines, 2020). As wildfires in the Siberian boreal forest are predominantly considered low-intensity surface fires (de Groot et al., 2013, Rogers et al., 2015), the potential of the resulting convection to transport large charcoal particles is probably limited compared to high-intensity crown fires. […] Even though some extreme fires may well surpass this estimate and, occasionally, small fires within might fail to contribute sufficient amounts of charcoal, identified fire episodes in the charcoal record should still be biased towards fires closer to the lake, especially when they consist of predominantly large charcoal particles (Conedera et al., 2009; Whitlock and Larsen, 2001)."

L 412-414: This assumption of the spatial scale reflected by the charcoal records would be much more useful if it came before presenting the FRI information – the meaning of FRI (and mean FRI) is contingent upon the spatial scale reflected or integrated across. Given the circumference of the lake, what does this translate to in terms of km^2? That's the key piece of information.

Thank you for pointing out how we could clarify the estimated charcoal source area. We agree that this is essential for all of the following discussion, which is why we will add an estimate in square km in its beginning as described above: "We therefore assume a charcoal source area between few hundred metres directly around the lake for low-intensity fires (Conedera et al., 2009) and increasing distance of up to several kilometres for more intense fires producing stronger convection, resulting in a total source area estimate of up to c. 100 km²."

L 441: Could this "contrast" between the current study and others, to some extent, reflect the differences in temporal scale? The current study is "only" 2000 yr long, whereas several of the studies cited span much longer time periods. Mechanisms for vegetation change vary over these different scales.

We agree, this is a good point and likely to have an effect on such comparisons! We will acknowledge that by adding a sentence in L442 in the revised manuscript: "A reason for this might be that studies on longer timescales capture long-term concurrent trends in both fire and vegetation that are not observable in the last c. 2000 yrs alone (e.g. glacial to interglacial changes in vegetation distribution and temperature; see Marlon et al., 2013)."

L 443-444: And...the very large spatial footprint integrated by the pollen record in this lake is key here. A clear pollen signal would require persistent vegetation change over a large area.

Correct. This is what we wanted to imply in L450 stating the pollen source area, but we feel that this was not done sufficiently. Thus, we will expand this aspect starting in L449: "In addition to

differences in proxy source area and taphonomy between macroscopic charcoal and pollen grains, a variety of factors likely obscures traces of potential fire impacts: surface fires in the deciduous forests in central Yakutia mostly result in the elimination of only a share of the affected tree population depending on fire intensity (Matveev and Usoltzev, 1996). This might not be enough to leave behind a clear mark in the pollen record, which also covers a source area that is probably way larger than the area affected by fire. Herbs or shrubs, on the other hand, may recover too quickly for changes to be detected in our record with a median temporal sampling resolution of 6 yrs. Any potential mixing processes and the residence time of pollen grains before settling in the lake sediment may further diminish visibility of a fire impact […]"

L 467-468: Doesn't this also directly apply to comparing the fire history record reconstructed here (i.e., one site, with a chronology subject to 14C reservoir effects) to any proxy with a well-constrained chronology?

From our point of view, such comparisons might hint towards the general viability of our reconstruction, although the constraints you mention need to be kept in mind. For instance, phase 3 with generally lower CHAR fits the timing of the LIA as indicated by other studies from Siberia and is also near the well-constrained top part of the chronology. For the lower half of the record, on the other hand, it is stated that comparability might be hampered by the assumptions behind our chronology (L485), thereby acknowledging that the problem might well lie within the present record instead of some other forcing yet to be recognized. We would also like to include the limiting factor of only having one site for now, by revising the last sentence of this paragraph (L487): "Yet, an impact of colder Arctic temperatures on the reconstructed low fire activity in phase 2 at Lake Khamra seems probable. Since we can currently rely only on this one record, more palaeoclimatic reconstructions and high-resolution charcoal records from the region could greatly improve the validity of such inferred links between climate and the fire regime."

L 543: Some citations would help identify the other studies noted here.

You are right, and this was mainly resembling global synthesis studies (e.g. Marlon et al., 2013), however, due to differences in temporal resolution and our underlying uncertainties we decided to drop the first part of this sentence. In the updated manuscript, L543 will read: "Recent levels of charcoal accumulation at Lake Khamra are not unprecedented within the last two millennia."

**Figure 1:** Classic CHAR comparing the **alternative local threshold (red)** to the global threshold (orange). Vertical dashed lines mark the different phases of the fire regime. (a) Classic CHAR peak component (dark grey bars = signal, light grey bars = noise, colored lines = threshold versions). (b) SNI after Kelly et al. (2011) (black horizontal line = SNI cutoff value of 3). (c) Classic CHAR sum (black line = interpolated CHAR, blue line = LOESS representing the CHAR background component, red/orange marks = fire episodes for local and global threshold versions, respectively, with color = fire episodes with SNI >3, in grey = fire episodes with SNI <3).

[Figure]

**Figure 2:** Comparison of **revised robust CHAR (including aggregation of three consecutive samples)** with its current version. Vertical dashed lines mark the different phases of the fire regime. (a) Classic CHAR peak component (dark grey bars = signal, light grey bars = noise, dashed horizontal line = threshold). (b) Current version of robust CHAR. (c) Revised robust CHAR. For (b) and (c): black line = median, grey area = interquartile range.

[Figure]

**References mentioned in this response:**

Bird, M.: Radiocarbon dating: charcoal, in Encyclopedia of Quaternary Science, edited by S. A. Elias and C. J. Mock, pp. 2950–2958, Elsevier, Amsterdam, https://doi.org/10.1016/B978-0-444-53643-3.00047-9, 2013.

Biskaborn, B. K., Herzschuh, U., Bolshiyanov, D., Savelieva, L., Zibulski, R. and Diekmann, B.: Late Holocene thermokarst variability inferred from diatoms in a lake sediment record from the Lena Delta, Siberian Arctic, J Paleolimnol, 49, 155–170, https://doi.org/10.1007/s10933-012-9650-1, 2013.

Biskaborn, B. K., Herzschuh, U., Bolshiyanov, D., Savelieva, L. and Diekmann, B.: Environmental variability in northeastern Siberia during the last ~13,300yr inferred from lake diatoms and sediment–geochemical parameters, Palaeogeogr. Palaeoclimatol. Palaeoecol., 329–330, 22–36, https://doi.org/10.1016/j.palaeo.2012.02.003, 2012.

Churakova Sidorova, O. V., Corona, C., Fonti, M. V., Guillet, S., Saurer, M., Siegwolf, R. T. W., Stoffel, M. and Vaganov, E. A.: Recent atmospheric drying in Siberia is not unprecedented over the last 1,500 years, Scientific Reports, 10(1), 15024, https://doi.org/10.1038/s41598-020-71656-w, 2020.

Clark, J. S., Lynch, J., Stocks, B. J. and Goldammer, J. G.: Relationships between charcoal particles in air and sediments in west-central Siberia, The Holocene, 8(1), 19–29, https://doi.org/10.1191/095968398672501165, 1998.

Conedera, M., Tinner, W., Neff, C., Meurer, M., Dickens, A. F. and Krebs, P.: Reconstructing past fire regimes: methods, applications, and relevance to fire management and conservation, Quat. Sci. Rev., 28(5), 555–576, https://doi.org/10.1016/j.quascirev.2008.11.005, 2009.

Dietze, E., Brykała, D., Schreuder, L. T., Jażdżewski, K., Blarquez, O., Brauer, A., Dietze, M., Obremska, M., Ott, F., Pieńczewska, A., Schouten, S., Hopmans, E. C. and Słowiński, M.: Human-induced fire regime shifts during 19th century industrialization: A robust fire regime reconstruction using northern Polish lake sediments, PLOS ONE, 14(9), e0222011, https://doi.org/10.1371/journal.pone.0222011, 2019.

Feurdean, A., Gałka, M., Florescu, G., Diaconu, A.-C., Tanţău, I., Kirpotin, S. and Hutchinson, S. M.: 2000 years of variability in hydroclimate and carbon accumulation in western Siberia and the relationship with large-scale atmospheric circulation: A multi-proxy peat record, Quaternary Science Reviews, 226, 105948, https://doi.org/10.1016/j.quascirev.2019.105948, 2019.

Fuchs, M., Bolshiyanov, D., Grigoriev, M., Morgenstern, A., Pestryakova, L., Tsibizov, L. and Dill, A.: Russian-German Cooperation: Expeditions to Siberia in 2019, Rep. Polar Mar. Res., https://doi.org/10.48433/BzPM_0749_2021, 2021

Gaglioti, B. V., Mann, D. H., Jones, B. M., Pohlman, J. W., Kunz, M. L. and Wooller, M. J.: Radiocarbon age-offsets in an arctic lake reveal the long-term response of permafrost carbon to climate change, Journal of Geophysical Research: Biogeosciences, 119(8), 1630–1651, https://doi.org/10.1002/2014JG002688, 2014.

[revised manuscript text omitted]

---

## Author Comment (AC2) · 19 Feb 2021

**Author response to RC2 by Angelica Feurdean**

Dear Angelica Feurdean,

we are happy to receive your review of our manuscript and wish to thank you for your time and effort. We welcome the feedback you provided, expanding especially our interpretation of the pollen record in important ways. Please find below your original comments in black, and our author response in green:

The manuscript by Glückler et al., is timely executed study of past wildfire dynamics and associated drivers in Eastern Siberia. The manuscript is based on high quality data and statistical analysis, is clearly written and well referenced. The study concluded that, at this temporal scale, it is climate and human impact, rather than vegetation driving the past millennial to centennial changes in wildfire activity.

Thank you for this assessment!

One of my main concern is the approach to the vegetation. The study is conducted in Larix (larch) dominated forests of Siberia. Larix is notoriously underestimated in the pollen records, and this should make it difficult to accurately reconstruct its past dynamics. Plant macrofossil analysis, mostly abundant needles (deciduous conifer tree), could help constrain its past dynamics, however, this is appropriately done in small basins or cores close to the lake margin, which is not the case of this site. The pollen record presented in this study neither show that Larix was an abundant taxon nor that its proportion have changed in the past, which likely highlights the problems above. I suggest adding a few lines acknowledging the problem of reconstructed past Larix dynamics based on pollen.

Thank you for raising this important issue. The Lake Khamra sediments did not provide many plant macrofossils and, as you mention, is not a perfect archive for the purpose of such analysis (also considering the suggested likelihood of prolonged permafrost-related terrestrial residence of macrofossils, as discussed for the chronology in the manuscript). We will update the manuscript to clearly acknowledgement the issue of *Larix* pollen dynamics in the discussion part for vegetation by adding the following starting in L455: "Furthermore, reconstructions of *Larix* dynamics based on fossil pollen are affected by a limited pollen dispersal distance of larch trees, as well as poor preservation of their pollen grains (Müller et al., 2010). This can lead to an underestimation of *Larix* in fossil pollen records (Edwards et al., 2000) and thus complicate the evaluation of larch tree dynamics. A comparably low share of fossil *Larix* pollen at Lake Khamra, despite *Larix gmelinii* being a predominant tree taxon within the study area, may well reflect that issue. This indicates how future studies, aiming at specifically comparing past fire regime changes with *Larix* population dynamics, may benefit from including plant macrofossil analysis, if possible (e.g. Birks, 2001; Stähli et al., 2006). Due to a lack of macrofossils in the Lake Khamra sediment core and their suggested prolonged terrestrial residence time, as implied by the chronology, this was not an option in the present study. These factors, together with a remaining ambiguity in morphotype classification, likely explain the rather weak correlations of pollen and charcoal records."

The other concerns on the chronology and charcoal peak analysis have been highlighted by the other reviewer (P. Higuera).

Please refer to our author response to review comment 1 to see our answers and improvements made in these regards.

Specific comments:

l. 63-63 Barhoumi et al.2019 study lies in European Russia not in Siberia, please correct

This will be corrected in the revised manuscript, by adding in L62/63: "[…] allowed the assessment of fire return intervals in boreal European Russia and western Siberian evergreen forests, revealing […]"

l.100 Note here that Larix gmelinii as one of the main tree taxa

This was probably meant to be related to L103 following; the sentence will be expanded as follows: "[…] forest consisting predominantly of *Larix gmelinii*, together with *Pinus sylvestris*, […]"

l.177 What do you mean by this? You broke the charr particles with a needle?

These preparatory needles were mainly used as a size reference. However, it turned out that they could also well be used to provide some "haptic" feedback when touching or pushing particles. This was helpful in some instances to differentiate between type B particles (those that have a brownish-black color instead of being uniformly black, but still solid pieces) or partially charred plant material (type Z), which will slightly bend when being touched by the needle. Breaking charcoal particles would render the re-counting of samples useless and was therefore avoided. To clarify this, we added to the sentence beginning in L175: "These needles also allowed the careful and non-destructive evaluation of the flexibility of particles of unknown origin, since […]".

Palynological analysis l. 191-194, Ok, but it would be useful to also state the resolution of pollen samples

We agree and will therefore move the statement about the resolution of charcoal samples, expanded with the timespan covered by the pollen samples specifically, to the results section in L280, where we think this information fits well: "Mean sampling resolution of the whole record is 7.1 ± 4.1 yrs (max: 27, min: 3), including the pollen samples, which on themselves cover on average 8.9 ± 3.8 yrs (max: 25, min: 4)."

l. 200 Which subsequent analysis? How were the % calculated?

The sentence starting in L199 will be expanded for clarification: "For subsequent statistical analyses, relative frequencies of individual pollen taxa were calculated from the sum of terrestrial pollen. Spore, algae and non-pollen palynomorph percentages are based on the sum of pollen plus either spores, algae or non-pollen palynomorphs, respectively (Andreev et al., 2020)."

l. 240 Please state what exactly was desired with the correlation between charcoal and pollen (not vegetation), and which pollen types were chosen and why. I got the feeling that the results from pollen record are minimized.

We will clarify the reason for the correlation test by changing the sentence beginning in L241 as follows: "Assuming that the various charcoal morphotypes were formed by different types of vegetation burning, we would expect an increase of a specific morphotype to coincide with changes in the distribution of some plant types in the vegetation composition, also represented by the pollen spectra. To explore this hypothesis, we applied a correlation test using Kendall's τ (package "psych"; Revelle, 2020) to clr-transformed relative distributions of pollen groups that were expected to be impacted by wildfires (pollen sums of arboreal, non-arboreal, deciduous and evergreen taxa, respectively) and charcoal classes following Dietze et al. (2020)." By stating the reason behind the correlations more clearly, the following discussion of the results is emphasized.

L.320 What was the propose on running CHAR on separate grain size and morphotypes? This is not stated in the methods.

We totally agree and will update the methods section to feature the following sentence in L238: "This was done in order to assess in detail their individual contribution to the sum of all particles, the way they capture a fire signal to see if different charcoal groups represent different types of fires, potential relationships between charcoal particle size and source area, and whether certain charcoal types represent varying fuel types over time."

l.325 Is there a difference between angular S and B morphotypes?

They possess the same overall features (angular shape, showing a charred surface structure), but in contrast to the jet-black type S particles, those of type B show brownish-black colors as well. This is based on the classification scheme by Enache and Cumming (2007). To clarify this difference, the sentence starting in L324 will be expanded as follows: "The most prevalent morphotypes present in the sediment are […], S (angular/black, 20.6%), and B (angular/brownish-black, 7.2%), with all others […]".

l.334 Do you mean similar pattern for all charcoal morphotypes?

If this relates to the sentence in L330, then yes! We will clarify this part as follows: "The three charcoal morphotype groups show a similar temporal pattern for their background and peak component distributions […]. However, when assessed individually, large particles have a generally lower variability than the other size classes, whereas the variability of irregular morphotypes is higher than that of elongated or angular particles (see Supplement)."

3.3 Vegetation history. I suggest adding the pollen diagram into the main paper. Would it make sense / increase visibility, to use continuous lines i.e, curves instead of bars to show trends in the pollen record? The past trends in vegetation are described in just two lines 353-355, an expansion of this is needed. In the pollen diagram (A1) there are two zones and at minimum the composition /differences between the 2 should be highlighted.

Thank you for your recommendation! We agree that added lines improve visibility of changes through time, while also keeping the bars gives clear indication of the position of samples involved. Please find the updated pollen diagram (Fig. 1) below, which we also expanded with an age axis and will now include in the results section directly.

To expand the description of past vegetation and the pollen zones, we will re-phrase the paragraph starting in L347 as follows: "The pollen and NPP record, covering the whole sediment core and reaching back c. 2350 yrs, generally indicates a relatively stable vegetation composition (Fig. 5). The dominant arboreal pollen (AP) types comprise most of the pollen spectra (average ratio of AP:NAP = 8.3:1) and include the trees and shrubs recorded around the lake. In descending order, regarding their share of the pollen sum, these are *Pinus*, *Betula*, *Picea*, *Abies*, *Alnus*, and *Larix*, with smaller amounts of *Salix*, *Juniperus*, and *Populus*. Non-arboreal pollen (NAP) types are predominantly represented by Cyperaceae, followed by less abundant Poaceae, Ericales, and *Artemisia*. Despite similar general palynomorph distributions, pollen assemblages are separated into two subzones, with the upper subzone (Ib) seeing intervals of increased variability in the shares of some tree pollen and Cyperaceaea (around 10 and 120 cm depth, corresponding to c. 1950 and 700 CE, respectively). The lower subzone (Ia) demonstrates generally lower shares of *Abies* and Cyperaceae pollen."

l. 347-355 I am confused by this statement, why is now Larix listed last? Additionally, the tree pollen composition may reflect that of the surrounding forest, but not the proportion. For ex Larix is one of the dominant taxa in the forest presently (according to the introduction and study area), however it was only found with scarred pollen grains.

We agree that the sentence beginning in L348 can be confusing as to whether is relates to modern vegetation or the pollen record. To clarify this order, we will add: "In descending order, regarding their share of the pollen sum, these are […]". By clearly stating that this is the list of shares in the pollen spectrum in descending order, the issue of underestimated *Larix* pollen is also emphasized, as it will be discussed shortly thereafter based on your previous comments.

l. 393 Barhoumi et al.2019 not in west Siberia

This will be corrected in the updated manuscript by adding in L392: "[…] lies within the range of the few comparable studies in boreal European Russia or western Siberia."

l. 405 Agree but this needs to be stated earlier in the methods and results.

To state the relationship of particle size and source area earlier, we will move this information to L322 in the results section: "When assessed individually, more fire episodes are identified for smaller particles than for larger particles (Table 2). This is expected, as smaller particles tend to have a larger source area, potentially incorporating more fire events into the signal (Conedera et al., 2009)." This will also be mentioned in the methods section (2.4), where we now improved the wording on our reasons for applying the statistics to individual charcoal classes as noted above.

l. 414 A few hundred meters is really little.

We agree that a charcoal source area of few hundred meters seems very small, however, there is some empirical evidence to include such a small area as a lower limit. For example, Ohlson and Treyterud (2000) used a grid of charcoal traps to capture particles from a fire of known location, and report a very high spatial resolution in the meter range for locally deposited large (> 500 μm) particles. Since we know from satellite imagery that fires burned directly at the lake's shore in 2006/7 (see Fig. 1b in the manuscript), we would therefore use this as a lower boundary of our source area. To clarify that this is considered to be a lower limit, we will re-phrase this part in the revised version of the discussion: "We therefore assume a charcoal source area between few hundred metres directly around the lake for low-intensity fires (Conedera et al., 2009) and increasing distance of up to several kilometres for more intense fires producing stronger convection, resulting in a total source area estimate of up to c. 100 km²."

l. 419 How then?

If this is related to the way we expect charcoal particles reaching the lake, we suggest that it is mainly via primary input through the air. We will include this important piece of information as follows: "This might indicate that morphotype distribution within the record is not controlled by potential filtering effects of secondary charcoal transport, but rather by the type of biomass burning. This is also implied by the predominantly primary charcoal input through the air due to the densely vegetated surrounding slopes, and mirrors the stable vegetation composition seen in the pollen record."

l. 425. I am a bit confused here. To which letter /type do the elongated type belong? I believe that burning graminoids would produce elongated charr particles also in Siberia, judging from other studies on the L:W ratio. However, there are others fuel types that have elongated morphologies.

We agree that the possibility of elongated charcoal from graminoids exits also in boreal Siberia, depending on the local vegetation. To clarify, we will update this part starting in L423: "Pereboom et al. (2020) found elongated charcoal particles after experimentally burning tundra graminoids, potentially hinting at the origin of the many elongated type F particles at Lake Khamra. However,

these type F particles quite closely match the appearance of charred *Picea* needles reported by Mustaphi and Pisaric (2014)."

4.2.1 Vegetation l. 435. Apart of the problem of large site and footprint on pollen record, please think whether biases with Larix pollen could have also 'falsely' contributed to this monotony. Are there other pollen diagrams in the region to document how vegetation composition varied regionally?

We agree that this is an important factor here. We will include the issue of underestimated *Larix* pollen in the revised discussion of the vegetation history, as laid out above. To the best of our knowledge the closest pollen record is located c. 600 km north-east of Lake Khamra, according to entries of the Neotoma database (www.neotomadb.org, accessed Jan 11th 2021), and therefore already within the predominantly deciduous forest as opposed to the more mixed evergreen/deciduous forest at our site. While this lack of well-comparable data is unfortunate, we hope that this study is able to contribute some first insight for future research to build upon.

l. 448 increased proportion of…

The sentence starting in L447 will be updated: "An increased proportion of evergreen trees might enable more intense crown fires."

l. 450 do you imply that pollen is more problematic than charcoal?

No, we rather wanted to put emphasis on the differences between the two proxies which might hamper their direct comparison as a way to test our hypotheses. We will clarify this by changing the sentence as follows: "In addition to differences in proxy source area and taphonomy between macroscopic charcoal and pollen grains, a variety of factors likely obscures traces of potential fire impacts: […]"

l.478 which? Please state the age range?

We will clarify the age we are referring to in the sentence starting in L476 by adding: "The following onset of increased fire frequency in phase 4 (c. 1750 CE onwards) is concurrent with a gradual increase in Arctic temperatures during the last two centuries […]"

l. 480 you may want to look at the moisture record https://doi.org/10.1016/j.quascirev.2019.105948

Thank you for drawing our attention to this study! Seeing that climatic periods such as the MCA and LIA could be identified provides further evidence for their manifestation in Siberia, and the reconstruction of wet and dry periods is a valuable addition. We will include this study in the discussion of the climatic impact (4.2.2) in the revised manuscript.

l. 550 given the large lake size, could charcoal input over time have been affected by different locations of fire in the catchment and the subsequent charcoal delivery into the lake?

Yes, unlike fire scar records from tree rings, we capture fires from a larger region. Fires closer to the lake likely lead to larger amounts of charcoal being deposited in its sediment, thus suspected to be responsible for outstanding peaks within the charcoal record. By retrieving the sediment core from the deepest point of the lake, which is suspected to be the terminal destination of sediment surface transportation pathways, we suspect to capture the highest possible amount of deposited charcoal. As discussed before, we expect the amount of secondary input via surface runoff to be limited by dense vegetation and gentle slopes within the catchment area. However, together with a revised discussion of lake size effects we will also acknowledge that in combination with a surface fire regime, some fires may be missed in our record: "Even though some extreme fires may well surpass this estimate and, occasionally, small fires within might fail to contribute sufficient amounts of

charcoal, identified fire episodes in the charcoal record should still be biased towards fires closer to the lake, especially when they consist of predominantly large charcoal particles (Conedera et al., 2009; Whitlock and Larsen, 2001)."

**Figure 1:** Revised pollen percentage diagram. Pollen and non-pollen palynomorphs from sediment core EN18232-3 at Lake Khamra (dots represent pollen taxa <1%; A = algae; Z = invertebrate remains; horizontal dashed line = separation in subzones Ia and Ib).

[Figure]

**References mentioned in this response:**

[revised manuscript text omitted]

---

## Author Comment (AC3) · 19 Feb 2021

Dear referee,

we would like to thank you for your time spent on reviewing our manuscript and value your honestly voiced concerns. Please find below your original review comments in black and our author responses in green:

This is a carefully prepared and well written manuscript. The topic is certainly important: fire records are needed from Siberia. Despite the detailed analyses, this is nevertheless a difficult site to interpret. I am not convinced by the results or their current interpretation for the following main reasons, which have been articulated in detail by the other reviewers.

Thank you for your assessment! We hope that our responses and improvements following your individual remarks below will clarify the results and provide an interpretation that better captures what our data can and cannot tell.

1) The chronology is very difficult. The offset of 1000 yr in the bulk sediment series may be approximately right, given there is carbonate bedrock in the vicinity. However, the mixed-up macrofossil dates suggest that material of different ages becomes incorporated into the sediment matrix, so why not the charcoal?

In contrast to the bulk sediment used for radiocarbon age dating, most of the matrix of which is thought to be either autochthonous or transported to the lake via erosional events or it's inflow stream before being reworked and finally accumulating on the lake bottom, charcoal particles follow an additional route of spreading through the environment via the air, lifted by the convection from fires themselves (primary input). We would expect the secondary input of charcoal particles from "old" fires (i.e. up to few tens of years ago) to be limited by fairly dense vegetation along the lake shore and the low angle of surrounding slopes. However, even such a lagged charcoal input would only apply to fires within the catchment area, whereas the suspected source area goes well beyond the catchment's boundary. Additionally, such secondary input would only lead to a "smoothing" of the charcoal chronology by increasing overall charcoal amounts in samples. Most dated macrofossils, on the other hand, are small structures from ground-dwelling plants that likely deposited after time-lagged secondary input only. For these reasons, the majority of charcoal particles are thought to be deposited soon after fires took place, instead of being a mere remnant of permafrost thaw like the other macrofossils potentially are.

As pointed out in another review, the charcoal could thus itself provide an opportunity to constrain the age offset throughout the sediment core. However, within the present study this is not feasible due to three main reasons: (i) small amounts of charcoal in present samples; (ii) scarce and valuable sediment material from an effortful helicopter expedition, preventing us from obtaining more material for that purpose; (iii) rigorous preparation steps of small charcoal particles as outlined by Bird (2013). Despite these difficulties, we will consider this promising approach for future studies.

2) The FRI's seem extraordinarily short. An average of 43, in a ~2000 yr series that has a quiescent period of ~600 yr is high, and when broken down into zones/phases, estimated FRI levels of 14 yr do not sound at all realistic. Nothing as short as this is reported from the region.

We agree that the mean FRI of phase 2 seems extraordinarily short at just 14 yrs. However, it is important to note how our charcoal record differs from reconstructions using other archives/proxies. We have not emphasized enough on this in our current version, but the revised manuscript will feature a clear description of three main reasons behind the short FRIs and differences to other

studies: (i) the large lake size allows to incorporate charcoal from a large source area, thus capturing more fires than more locally constrained studies at small lakes or with tree ring chronologies. This leads to (ii): we can interpret only fire episodes instead of individual fires, because of the large source area and a lower-intensity surface fire regime (i.e. a single peak of the charcoal record is not an individual fire, but rather a phase of increase fire activity). (iii) Our statistical approach considers adjacent peaks above a threshold as individual fire episodes, where conventionally, only the highest peak is considered. We assume that the ability of a single fire to create multiple adjacent outstanding peaks is limited at this study site due to quick recovery of dense vegetation on low angle slopes, limiting secondary input, and sediment mixing processes likely not exceeding the sampling resolution of the charcoal record.

Furthermore, most existing studies utilize tree ring chronologies which might only record fires that did not kill a tree population at the same site, whereas the charcoal record includes also such higher-intensity fires across a larger region. Also, most of these studies (e.g. Kharuk et al., 2008, 2011) seem to be located further north, while fires are most frequent in central and southern Yakutia (see Ivanova 1996). It does seem like few degrees in latitude can have a striking impact on fire frequency, especially when combined with differences in stand-specific moisture conditions. For example, a mean FRI of 15 yrs was reported by Takahashi (2006) near Yakutsk, and similarly low values are found in Ivanova (1996) in central Yakutia, with the longest FRIs not exceeding 40–50 yrs on moist sites and only half of that on dry sites.

3) The relevance of calculating FRI's for different types of charcoal morphology is not explained and no convincing implications of doing this, or the results, are presented. Thus despite excellent detailed methodology, it would be difficult to draw much that is useful from this study. This study would be better presented as simpler types of time series and together with the pollen (as suggested by the other reviewers); it is far preferable to treat the data appropriately than to develop complex analyses that could well provide a misleading picture of events.

Applying our statistical approach not only to the sum of charcoal particles, but also to specific sub-categories of size classes or morphotypes, aims at evaluating how these different categories contribute to the overall charcoal record and differ from each other. The results are included in Fig. 5c, d of the manuscript, showing the peak frequencies and background components for the various size classes and morphotype groups, with more detailed individual diagrams available in the Supplement. We totally agree that the purpose of this has not been sufficiently stated, and we will add this information to the updated manuscript in L238: "This was done in order to assess in detail their individual contribution to the sum of all particles, the way they capture a fire signal to see if different charcoal groups represent different types of fires, potential relationships between charcoal particle size and source area, and whether certain charcoal types represent varying fuel types over time."
One relevant result of applying the statistics to the size classes individually is briefly stated in L322, which will be expanded as follows: "The three charcoal morphotype groups show a similar temporal pattern for their background and peak component distributions (Fig. 6c, d), mostly mirroring the decreased variability in the second half of the record as described for the sum of all particles. However, when assessed individually, large particles have a generally lower variability than the other size classes, whereas the variability of irregular morphotypes is higher than that of elongated or angular particles (see Supplement)." Since charcoal morphotypes are not yet well constrained in their interpretations, we feel that their inclusion within this high-resolution record and the separate application of our statistics is justified and might be of interest to future studies looking for comparisons.
In general, our analyses aim at incorporating uncertainties, which is the prime reason for including the robust CHAR approach, and we try to communicate those as best as we can. Thus, we agree that

the usage of certain terms (i.e. "FRI", "fire frequency") can be misleading here without proper definition, especially when put in comparison to studies using other proxies. For that reason, we will include a new paragraph at the beginning of the discussion to provide definitions of the peak component as "fire episodes", the expected charcoal source area and how, with these two key points of information, the term "FRI" is to be understood in the present study. In the end, this is, to the best of our knowledge, the first study working on a high-resolution macroscopic charcoal record from eastern Siberia, while also developing new statistical approaches to better capture uncertainties. Therefore, from our point of view, the present study provides a strongly needed foundation upon which further research will be able improve.

Other comments

L80 useful to mention any estimates of FRI here in description of region, seeing this is a fire study.

We agree and will add FRI estimates from studies set in Yakutia to the description of the study region in L110: "Wildfires most frequently occur in the central to southern regions of Yakutia, with varying stand-specific mean fire return intervals not exceeding 20–50 yrs at around 60°N, 120°E (Ivanova 1996) and 15 yrs near Yakutsk (Takahashi, 2006). Longer estimates, increasing with higher latitude, were found by Ponomarev et al. (2016) in central Siberia with 80 yrs at 62°N, 200 yrs at 66°N and 300 yrs at 71°N, with similar ranges reported by Kharuk et al. (2016)."

L187 need to clarify a bit more about the samples that had no char analysis; was it just the 11?

Unfortunately, 11 samples (all within the top 40 cm of the sediment core) were used for a destructive type of analysis test before morphotype classification could be applied. However, total charcoal concentrations and size class distributions of these samples were obtained beforehand. This will be clarified in the revised manuscript: "At that time, 11 samples distributed within the top 40 cm of the sediment core had already been used for other purposes and thus lack information on morphotype classification (total charcoal concentrations and size classes are available for all samples)."

L206 well established

Thank you!

L250 Between lines 250 and 263 there is repetition, this part needs re-writing. In general, the discussion of the radiocarbon dates is long and over-complicated. It would help the reader to place a statement about how mixed the radiocarbon dates are right up front and state the whole problem much more directly.

We think that this discussion is a necessary part of this paper as the chronology is the largest source of uncertainty, and it also serves as a case study on a prominent issue with paleoenvironmental studies set in permafrost regions. We believe issues of that kind should be thoroughly discussed, and not put as a side note. Many studies depend on such chronologies whilst not providing in detail the assumptions or potential issues behind them (see Lacourse and Gajewski, 2020), which by doing so might motivate our community to eventually find more effective solutions. We agree that the readability of this part could be improved, so we will try to streamline this discussion in the updated manuscript. To better introduce the reader, a sentence briefly summarizing the age dating outcome will be added in L252: "Bulk sediment [14]C ages indicate a rather linear chronology, with only the deepest two samples returning similar ages. In contrast to this, the macrofossils [14]C ages do not indicate a clear chronological pattern and are not in good agreement with the bulk sediment samples". In L261, the sentence part "which shows a recent surface age" will be deleted because of repetition.

L266 sentence beginning "(II) Macrofossil 14C ages...." Not quite sure what this means

The plant macrofossil dated at ~10.000 $^{14}$C BP in a surrounding sediment matrix that is likely only at ~100 yrs BP shows directly how the presence of "old carbon" can, when included in a bulk sediment sample, artificially lead to an older-than-expected age result. This sentence should emphasize that there is direct evidence for the presence of old carbon with the mixed macrofossil ages, apart from the age offset between surface $^{210}$Pb/$^{137}$Cs and $^{14}$C results. It will be rephrased to: "Macrofossil $^{14}$C ages older than the surrounding sediment matrix provide direct evidence for the potential influence of old carbon on bulk sediment samples at various depths (e.g. macrofossil age of 9902 ± 97 $^{14}$C yrs BP in sediment that dates back to only c. 100 yrs BP according to the parallel core's Pb/Cs age)."

L345 The relevance of the PCA is hard to see; more explanation in caption would help. Given the lack of impact of the morphology data (and Fig 4 only mentioned in results once), this could be omitted.

The PCA indicates that there is no clear clustering of charcoal morphotype or size classes with increasing core depth. Additionally, it visualizes correlations between the charcoal classes, showing how irregular type M particles are more closely associated with small particles. We think that these are relevant bits of information, as they also add to the discussion of weak morphotype correlations in L445-455. To further highlight the relevance of the PCA, we will add to L330: "Furthermore, the PCA indicates that there are rather weak grouping patterns of morphotype or size-class distributions in samples of increasing core depth and age, potentially reflecting a stable vegetation composition around the lake." Additionally, the PCA caption in L345 will be expanded with: "[…], with colored dots representing potential grouping patterns of charcoal assemblages with increasing age."

L400. "In general, FRIs increase with latitude due to lower incoming solar radiation, shorter fire seasons, and lower flammability of moist biomass (Kharuk,2016; Kharuk et al., 2011), which likely contributes to a relatively short mean FRI at Lake Khamra." Explain further? This site has a short FRI therefore...it is further south than other sites? The argument is not clear.

We argue that at Lake Khamra, located further South compared to many of the other cited studies, it is in that sense not unexpected to see shorter FRIs (in addition to differences in archive and proxy used, as noted above). To make this better understandable, we will re-phrase this sentence: "In general, fire frequency tends to increase with decreasing latitude due to higher solar radiation, longer fire seasons, and higher flammability of dry biomass (Ivanova, 1996; Kharuk, 2016; Kharuk et al., 2011), which likely contributes to a relatively short mean FRI at Lake Khamra when compared to studies set further north."

L 419 argument is a bit hard to follow in sentence beginning "However, the present "

In essence, the original argument by Enache and Cumming (2007) is that fragile morphotypes in lake sediment originate preliminary from primary input through the air, as they would likely be destroyed during longer distances of secondary input via surface runoff. When the catchment to lake-area ratio is large, the distance of potential surface runoff increases and would therefore "filter" secondary input of fragile morphotypes. However, even though Lake Khamra has a large catchment to lake-area ratio, the charcoal record is dominated by fragile particles. This is why we suspect other factors, like the type of biomass burning, to be mainly responsible for the morphotype distribution we see in the sediment. To improve the wording of this reasoning, the paragraph from L415-L421 is rephrased to: "Enache and Cumming (2007) explain how a large catchment to lake-area ratio might favour secondary deposition of compact/stable morphotypes, while fragile morphotypes are more prone to fragmentation during surface runoff and thus rather represent primary input through the air. However, the catchment to lake-area ratio at Lake Khamra (23:1), as well as the share of fragile charcoal particles (types F, M, and S alone make up >80%), are both comparably large. This might

indicate that morphotype distribution within the record is not controlled by potential filtering effects of secondary charcoal transport, but rather by the type of biomass burning. This is also implied by the predominantly primary charcoal input through the air due to the densely vegetated surrounding slopes, and mirrors the stable vegetation composition seen in the pollen record."

L459 there is low peak frequency during much of what might be thought of as the MO and higher one toward the end of the LIA, so there is not a very good fit to climate – this is over-interpreted, especially given the caveats provided and the difficult chronology

It is correct that the general timeframe of the MO/MCA (c. 950 – 1250 CE according to Mann et al. 2009) does not match the inferred fire activity from the present charcoal record. Even though this is acknowledged in L459, stating the underlying chronology as an additional issue, we agree that this part should be rephrased to better represent its foundation in our data. On the contrary, low CHAR during phase 3, in a time before industrialization and rapid population growth and with the stable vegetation composition, would lead us to suspect a cooler and/or wetter climate as the main driver. Seeing then how various other studies find evidence for the LIA during that time frame, we think it is not unlikely that this is captured by our record as well. We suggest the following revision of this paragraph:

"Although it has been demonstrated that the timing and extent of supposedly ubiquitous warmer or cooler climatic phases are in fact heterogeneous (Guiot et al., 2010), evidence for their occurrence in Siberia is seen in proxy studies (Churakova Sidorova et al., 2020; Feurdean et al., 2019; Kharuk et al., 2010; Osborn and Briffa, 2006), albeit less pronounced when it comes to vegetation response in the West Siberian Lowland (Philben et al., 2014). Neukom et al. (2019) show how such climatic periods arising from averaged reconstructions at many individual study sites are not spatially or temporally coherent on the global scale, and conclude that environmental reconstructions "should not be forced to fit into global narratives or epochs". This might be especially true for studies of a single site, using chronologies that have $^{14}$C reservoir effects. However, the low fire activity in the latter half of phase 3 (900–1750 CE) strikingly coincides with the Little Ice Age (LIA), when in many regions of the Northern Hemisphere a cooler climate prevailed from c. 1400 to 1700 CE. In contrast to this, high fire activity during phase 2 not matching the proposed timing of the warmer Medieval Climate Anomaly (MCA, c. 950 to 1250 CE) demonstrates the limitations of such comparisons based solely upon the $^{14}$C-dated segment of the charcoal record (estimates of LIA and MCA durations from Mann et al., 2009)."

**References mentioned in this response:**

Bird, M.: Radiocarbon dating: charcoal, in Encyclopedia of Quaternary Science, edited by S. A. Elias and C. J. Mock, pp. 2950–2958, Elsevier, Amsterdam, https://doi.org/10.1016/B978-0-444-53643-3.00047-9, 2013.

Churakova Sidorova, O. V., Corona, C., Fonti, M. V., Guillet, S., Saurer, M., Siegwolf, R. T. W., Stoffel, M. and Vaganov, E. A.: Recent atmospheric drying in Siberia is not unprecedented over the last 1,500 years, Scientific Reports, 10(1), 15024, https://doi.org/10.1038/s41598-020-71656-w, 2020.

Enache, M. D. and Cumming, B. F.: Charcoal morphotypes in lake sediments from British Columbia (Canada): an assessment of their utility for the reconstruction of past fire and precipitation, J Paleolimnol, 38(3), 347–363, https://doi.org/10.1007/s10933-006-9084-8, 2007.

Feurdean, A., Gałka, M., Florescu, G., Diaconu, A.-C., Tanţău, I., Kirpotin, S. and Hutchinson, S. M.: 2000 years of variability in hydroclimate and carbon accumulation in western Siberia and the

relationship with large-scale atmospheric circulation: A multi-proxy peat record, Quaternary Science Reviews, 226, 105948, https://doi.org/10.1016/j.quascirev.2019.105948, 2019.

Guiot, J., Corona, C. and Members, E.: Growing Season Temperatures in Europe and Climate Forcings Over the Past 1400 Years, PLOS ONE, 5(4), e9972, https://doi.org/10.1371/journal.pone.0009972, 2010.

Ivanova, G. A.: The Extreme Fire Season in the Central Taiga Forests of Yakutia, in Fire in Ecosystems of Boreal Eurasia, edited by J. G. Goldammer and V. V. Furyaev, pp. 260–270, Springer Netherlands, Dordrecht, 1996.

Kharuk, V. I.: Larch forests of Middle Siberia: long-term trends in fire return intervals, Reg Environ Change, 2389–2397, https://doi.org/10.1007/s10113-016-0964-9, 2016.

Kharuk, V. I., Ranson, K. J., Dvinskaya, M. L. and Im, S. T.: Wildfires in northern Siberian larch dominated communities, Environ. Res. Lett., 6(4), 045208, https://doi.org/10.1088/1748-9326/6/4/045208, 2011.

Kharuk, V. I., Im, S. T., Dvinskaya, M. L. and Ranson, K. J.: Climate-induced mountain tree-line evolution in southern Siberia, Scandinavian Journal of Forest Research, 25(5), 446–454, https://doi.org/10.1080/02827581.2010.509329, 2010.

Kharuk, V. I., Ranson, K. J. and Dvinskaya, M. L.: Wildfires dynamic in the larch dominance zone, Geophys. Res. Lett., 35(1), L01402, https://doi.org/10.1029/2007GL032291, 2008.

Lacourse, T. and Gajewski, K.: Current practices in building and reporting age-depth models, Quaternary Research, 96, 28–38, https://doi.org/10.1017/qua.2020.47, 2020.

Mann, M. E., Zhang, Z., Rutherford, S., Bradley, R. S., Hughes, M. K., Shindell, D., Ammann, C., Faluvegi, G. and Ni, F.: Global Signatures and Dynamical Origins of the Little Ice Age and Medieval Climate Anomaly, Science, 326(5957), 1256–1260, https://doi.org/10.1126/science.1177303, 2009.

Neukom, R., Steiger, N., Gómez-Navarro, J. J., Wang, J. and Werner, J. P.: No evidence for globally coherent warm and cold periods over the preindustrial Common Era, Nature, 571(7766), 550–554, https://doi.org/10.1038/s41586-019-1401-2, 2019.

Osborn, T. J. and Briffa, K. R.: The Spatial Extent of 20th-Century Warmth in the Context of the Past 1200 Years, Science, 311(5762), 841–844, https://doi.org/10.1126/science.1120514, 2006.

Philben, M., Kaiser, K. and Benner, R.: Biochemical evidence for minimal vegetation change in peatlands of the West Siberian Lowland during the Medieval Climate Anomaly and Little Ice Age, Journal of Geophysical Research: Biogeosciences, 119(5), 808–825, https://doi.org/10.1002/2013JG002396, 2014.

Ponomarev, E., Kharuk, V. and Ranson, K.: Wildfires Dynamics in Siberian Larch Forests, Forests, 7(12), 125, https://doi.org/10.3390/f7060125, 2016.

Takahashi, K.: Future perspectives of forest management in a Siberian permafrost area, in Symptom of Environmental Change in Siberian Permafrost Region, edited by Hatano, R. and Guggenberger, G., Hokkaido University Press, Sapporo, pp. 163-170, 2006.

---

## Author Comment (AC4) · 19 Feb 2021

Dear Daniel Gavin,

we appreciate your helpful comment about the robust CHAR approach used in our manuscript and your constructive thoughts on improvements! Please find below your original remarks in black, and our responses in green:

I wish to submit a short comment on one part of the paper regarding the robust charcoal method originally described in Dietz et al. 2019 in PlosOne.

Incorporating uncertainties into proxy records, including both the age uncertainty and the uncertainty of the proxy itself, is important especially when comparing periods within a core and when comparing sites. It is overdue to include uncertainty in the analysis of individual sediment charcoal records. So, it is great to see this extension of the methods from the 'ensemble' approach from Blarquez et al.

My comment addresses the resampling methods used for estimating the uncertainty of the sediment accumulation rates. The robust method uses the age estimate of each sample (described as a mean and sd, but it could also be a PDF from an age-depth model), and selects ages from that PDF. Ages are generated independently for all samples, and only ages in adjacent samples that are in chronological order are retained. This results in some very slow sedimentation rates. This is acknowledged in the 2019 paper: "A comparison showed that robust fluxes were smoothed, but underestimated absolute mean fluxes due to strongly overlapping pdfage of adjacent samples at 1 cm sample resolution. Hence, we averaged the raw proxy and age values of three adjacent samples before robust flux calculation."

I am not sure how the averaging as described makes the CHAR influx values more comparable to the original influx values. did you average three samples in nonoverlapping segments, thus increasing the age difference of adjacent composite samples? The presented robust CHAR values are small compared to the raw data. I do not see this averaging step in the supplied code.

Thank you for drawing our attention to this important part of the method! Indeed, as you assumed, we did not apply an averaging of multiple adjacent samples in the current version of the manuscript. We have now run additional tests of robust CHAR parameters with different resampling intervals on our charcoal record, and concluded that the influence on the general trends is limited, whereas overall CHAR values are increased the more samples are aggregated. However, it rather seems that the different magnitudes of age uncertainties of the $^{14}$C compared to $^{210}$Pb/$^{137}$Cs ages of the record have a much higher impact on the resulting trend than the resampling. The current approach of scaling both to more comparable dimensions of 1σ and 2σ ranges, respectively, seems reasonable to us, which is why we suggest sticking to it. However, including an averaging of adjacent samples makes sense based on the high temporal resolution of the record and leads to CHAR magnitudes more comparable to the classic approach. For this reason, we will exchange the current version of robust CHAR in Fig. 3 with a slightly different one in the revised manuscript, which includes averaging over three adjacent samples following Dietze et al. (2019) (see Fig. 1 below). Some peaks of classic CHAR, for example in the latter half of phase 2, are still not mirrored in robust CHAR. However, we would expect to only see a conservative estimate in robust CHAR, meaning that only phases of increased CHAR that stand out even with the relatively large added uncertainties remain visible. Apart from that, the revised diagrams do provide a better fit to classic CHAR, especially in phase 4. Furthermore, we included this resampling step in the revised R script. It is now possible to easily

choose at the beginning of the script whether a sample aggregation should be included, and across how many adjacent samples the averaging should take place. (script is available at https://github.com/rglueckler/CharcoalFireReconstructionR/tree/revised).

When the PDFs of adjacent samples are overlapping (<2 sd), the median age difference of the simulated ages is greater than the difference in the mean ages of the best-fit age-depth relationship. This is demonstrated in the attached figure. I think such small age differences occur in the majority of Holocene sediment records. The net effect is that as a core varies in sedimentation rate, the simulated sedimentation rate will have an increasing effect from the overlapping PDFs as the sedimentation rate decreases. This results in different effects of the analysis occurring in different parts of the same core. Variability in simulated sedimentation rates will not vary directly with the variation in the sedimentation of the best-fit age-depth model.

You are right. The difference is related to using only positive values for unit deposition times (the inverse of the sedimentation rate) and for the flux density distributions. We have now preserved the whole PDFs of unit deposition times in part 1 and of the proxy-flux calculations in part 2 of the robust CHAR script, as at these stages it is not necessary to remove the negative part of the PDFs (i.e. two dispensable lines of code in the robust CHAR function). Hence, only when we calculate the empiric flux density function we keep the positive values of the flux PDF. This results in slightly higher overall influxes but only in a small order of magnitude. The more pronounced effect is related to the relation of age uncertainties and sample resolution, as large age uncertainties will also lead to a wide spread of sedimentation rates for adjacent samples. We can account for this to a certain limit with the added sample aggregation.

An alternative approach to simulating sedimentation rates: use the output from bacon or clam, which saves many simulated runs of age-depth relationships. These can be used directly in the robust char calculations. The advantage here is that the simulated age-depth relationships preserves the monotonic age-depth pattern. The necessary ages for using this approach are in objects saved by the bacon and clam programs. (objects called info, dat, or chron). Clam and bacon can apply age uncertainties to proxy records directly. However, you have more flexibility by using the set of simulated age-depth relationships.

We appreciate this suggested alternative approach! It seems advantageous to use the age-depth model output directly to include the full range of multi-modal age distributions per depth. However, after some testing we think this is not a trivial task and requires more fine-tuning of our current method in order to be fully realized, which is beyond the scope of our present study. We will therefore work on implementing this approach in upcoming studies to further improve the robust CHAR methodology!

**Figure 1:** Comparison of **revised robust CHAR (including aggregation of three consecutive samples)** with its current version. Vertical dashed lines mark the different phases of the fire regime. (a) Classic CHAR peak component (dark grey bars = signal, light grey bars = noise, dashed horizontal line = threshold). (b) Current version of robust CHAR. (c) Revised robust CHAR. For (b) and (c): black line = median, grey area = interquartile range.

[Figure]

**References mentioned in this response:**

Dietze, E., Brykała, D., Schreuder, L. T., Jażdżewski, K., Blarquez, O., Brauer, A., Dietze, M., Obremska, M., Ott, F., Pieńczewska, A., Schouten, S., Hopmans, E. C. and Słowiński, M.: Human-induced fire regime shifts during 19th century industrialization: A robust fire regime reconstruction using northern Polish lake sediments, PLOS ONE, 14(9), 1–20, https://doi.org/10.1371/journal.pone.0222011, 2019.

---

## Author Comment (AC5) · 19 Feb 2021

Dear Sandy Harrison,

thank you for summarizing the issues you would have us pay special attention to. We fully agree that the review comments are very constructive and provide well-founded advice to improve the manuscript, and help to better communicate the underlying uncertainties constraining the interpretation of this study. Regarding the listed issues, you can find the most detailed answers in our responses to the following individual review

comments:

For 1) the reliability of the chronology and the impact of chronological uncertainties on the conclusions: RC1, as well as RC3

For 2) the appropriateness of the calculated fire return intervals: RC1

For 3) the re-sampling methodology and its impact on the results: SC1

For 4) how/why the fire peak analysis differs from conventional methods and the impact of this on the reliability of the reconstructions: RC1

For 5) the vegetation data used and particularly the degree to which underrepresentation of key pollen types is likely to have impacted your conclusions: RC2

With best regards

Ramesh Glückler